# Control of lymphatic pacemaking and pumping by mechanobiological signals

M. J. Davis[1] 🆔 and C. D. Bertram[2] 🆔

[1]*Department of Medical Pharmacology & Physiology, University of Missouri, Columbia, MO, USA*
[2]*School of Mathematics & Statistics, University of Sydney, Sydney, NSW, Australia*

Handling Editors: Kim Barrett & Bernard Drumm

The peer review history is available in the Supporting Information section of this article (https://doi.org/10.1113/JP288477#support-information-section).

**Abstract figure legend** Depiction of how flow (via shear stress on lymphatic endothelial cells (LECs)) and pressure (via stretch of lymphatic muscle cells (LMCs)) might modulate the amplitude and frequency of spontaneous lymphatic contractions (and thereby active lymph transport). Imposed flow causes sufficient NO production to dilate the vessel and completely inhibit spontaneous contractions. Elevated pressure/stretch causes increased contraction frequency with some compromise in contraction amplitude. Pulsatile shear stress (SS) produces an intermediate effect, slowing frequency to allow more time for diastolic filling, which increases contraction amplitude. Also depicted is a prediction that flow/pulsatile SS favours antegrade conduction of contraction waves, whereas pressure/stretch favours retrograde conduction, which is not supported by our experimental results.

Michael J. Davis is a Curators' Distinguished Professor in the Department of Pharmacology and Physiology at the University of Missouri School of Medicine. His research interests include lymphatic biology, smooth muscle electrophysiology and vascular mechanotransduction. **Christopher D. Bertram** is an Honorary Professor in the School of Mathematics and Statistics, Faculty of Science, at the University of Sydney. His research interests include modelling the lymphatic system, flow-induced oscillations, collapsible tubes and spinal cord hydrodynamics.

**Abstract** The spontaneous, phasic contractions of collecting lymphatic vessels are critical for lymph transport and interstitial fluid homeostasis. Phasic contractions are initiated by action potentials in lymphatic muscle and conduct along the vessel to trigger contraction waves. Contractions are regulated by pressure and shear stress (SS), but only limited aspects of that regulation are understood. Numerical models predict that pressure promotes retrograde propagation of contraction waves, whereas nitric oxide (NO) production associated with phasic contractions (pulsatile NO) promotes antegrade conduction and extends the pressure range over which contractions propel lymph. These predictions were tested using 3−4-valve segments of rat mesenteric lymphatic vessels using pressure myography and protocols that imposed forward flow, elevated inflow pressure ($P_{in}$) or elevated outflow pressure ($P_{out}$), each with/without intact NO signalling. NO bioavailability and flow-induced responses were enhanced by L-arginine supplementation. Spatiotemporal maps generated from video images were used to quantify the direction and extent of contraction wave conduction. Our results show that (1) contraction waves are normally biased towards retrograde conduction at equal $P_{in}/P_{out}$ levels. (2) $P_{in}$ elevation promotes antegrade conduction, whereas $P_{out}$ elevation promotes retrograde conduction. (3) Imposed flow is inhibitory, reducing contraction amplitude and frequency and limiting the extent of contraction wave conduction without a significant effect on conduction direction. (4) Pulsatile NO does not significantly influence the conduction direction or extend the pressure range over which spontaneous contractions occur. Our findings support the idea that pressure is the dominant regulator of lymphatic pacemaking and pumping, with pulsatile NO having only minimal influence.

(Received 9 January 2025; accepted after revision 23 April 2025; first published online 4 June 2025)

**Corresponding author** Michael J. Davis: Department of Medical Pharmacology & Physiology, University of Missouri School of Medicine, One Hospital Drive, MA415 Medical Sciences Building, Columbia, MO, USA. Email: davismj@health.missouri.edu

**Key points**

- The degree to which spontaneous, phasic contractions of lymphatic collecting vessels are regulated by pressure and shear stress is not fully understood.
- Numeric models predict that nitric oxide (NO) production associated with phasic contractions (pulsatile NO) promotes antegrade conduction of contraction waves, whereas pressure elevation promotes retrograde conduction; pulsatile NO production is also thought to extend the pressure range over which phasic contractions occur.
- *Ex vivo* methods were used to control pressure/flow in 3−4 valve segments of collecting lymphatics from rat mesentery, with preserved or inhibited NO signalling.
- The relatively long vessel segments limited the absolute levels of imposed flow/SS, so L-arginine supplementation was used to enhance NO bioavailability.
- Our findings support a scheme whereby pressure is by far the dominant mechanism determining the pacemaking site of lymphatic collectors, and challenge existing dogma about the importance of pulsatile NO production in regulating their behaviour.

## Introduction

The spontaneous contractions of collecting lymphatic vessels, in conjunction with periodically spaced one-way valves, comprise an active lymph pump system that facilitates the central transport of lymph. The pumping unit of this system, the 'lymphangion' (Mislin, 1961), is the segment of collecting vessel that includes two intraluminal valves. Lymphangions are encircled by one or more layers of spontaneously active lymphatic muscle cells (LMCs), and their contraction cycles are analogous to those of a cardiac chamber, with each 1−3 s-long cycle consisting of a rapid systolic ejection phase in which lymph is propelled through the outflow valve, and a slower diastolic relaxation phase in which the lymphangion refills through the inflow valve (Scallan et al., 2012). Contractions are initiated by action potentials in LMCs (Zawieja et al., 2024) and conduct as electrical activation waves at

speeds of ∼10 mm/s (Castorena-Gonzalez et al., 2018), with a variable degree of entrainment between adjacent lymphangions (Castorena-Gonzalez et al., 2018; Crowe et al., 1997; McHale & Meharg, 1992; Zawieja et al., 1993).

Like the arterial vasculature, the pumping activity of lymphatic collectors is sensitive to and regulated by mechanobiological forces. Interstitial fluid accumulation results in increased absorption of water, solute and protein by lymphatic capillary networks upstream of the collecting vessels. In the capillaries flap valves created by overlapping junctions between lymphatic endothelial cells (LECs) ensure net reabsorption (Petrova & Koh, 2020). Both capillary and collecting vessel networks are low-pressure systems with physiological operating ranges (in rodents) between 0 and 15 cmH$_2$O (Zweifach & Lipowsky, 1984). Collecting vessels are exquisitely sensitive to increases in intraluminal pressure, which expand the vessel wall and trigger increased activity of LMC ionic pacemakers to increase the frequency of spontaneous (phasic) contractions (Zawieja et al., 2024). The contraction frequency of a mouse popliteal lymphatic collector can change by 10-fold as pressure rises from 0.5 to 5 cmH$_2$O (Davis & Zawieja, 2025; Zawieja, Castorena-Gonzalez, To et al., 2018) – a regulatory mechanism referred to as pressure-induced chronotropy. Changes in pressure within the collecting vessel lumen also modulate the strength of phasic contractions, with amplitude increasing between 0.5 and 2 cmH$_2$O (in rodents) and declining at higher pressures (Scallan et al., 2012; Zawieja, Castorena-Gonzalez, To et al., 2018; Zhang et al., 2007), as dictated by the length/active tension relationship for striated and smooth muscle (Zhang et al., 2013).

Phasic lymphatic contractions are also regulated by shear stress (SS) but in a more complicated manner than in the arterial vasculature. In arteries increased flow/SS elicits vasodilatation through the production of diffusible endothelium-derived mediators, including nitric oxide (NO). In lymphatic collectors flow and SS have biphasic effects on spontaneous contractions. Fluctuations in SS occurring as a consequence of active lymphatic pumping are known to result in pulsatile NO production (Bohlen et al., 2009). These relatively modest and localized increases in NO result in an increased rate of diastolic relaxation, allowing time for more complete filling of the lymphangion, which in turn results in a slightly higher contraction amplitude of the subsequent contraction (Gasheva et al., 2006). Higher levels of flow result in higher levels of NO production, which inhibit both contraction amplitude and frequency, and lead to complete cessation of phasic contractions in the thoracic duct (TD) and more modest inhibitory effects on phasic contractions of peripheral collecting lymphatics (DuToit et al., 2024; Gashev et al., 2002, 2004, 2006; Quick et al., 2009; Scallan et al., 2012). In arteries the effects of NO

are supplemented by SS-induced endothelium-dependent hyperpolarization (EDH), which is conducted from endothelial cells to arterial smooth muscle cells through myo-endothelial gap junctions (Davis, Earley et al., 2023); in lymphatic collectors an EDH mechanism is not operative because LECs lack strong electrical coupling to LMCs (Castorena-Gonzalez et al., 2018; Hald et al., 2018; von der Weid et al., 2001). The responses of lymphatic collectors to flow are generally blocked by inhibitors of soluble guanylate cyclase or NO synthase (Gashev et al., 2002, 2006), with some exceptions (DuToit et al., 2024; Kornuta et al., 2015; Mukherjee et al., 2019; Nizamutdinova et al., 2014), suggesting that flow-induced responses are mediated primarily by NO.

Kunert et al. (2015) proposed an elegant numerical model to describe the interacting effects of flow and pressure on lymphatic pacemaking and pumping. The model postulates that individual lymphatic contractions are initiated by stretch or electrical pacemaker activation causing calcium ion entry and/or release (implicitly in LMCs) and terminated by NO production (implicitly from LECs) in response to the consequent flow-induced increase in wall SS. Because an LMC Ca$^{2+}$ increase necessarily precedes contraction, whereas NO production follows, these initiating and terminating events happen consecutively rather than cancelling each other out, and a repetitive cycle is set up. Based largely on the behaviour of mouse popliteal lymphatic vessels *in vivo* and the consequences of deleting eNOS or iNOS genetically, the model makes the following predictions (Kunert et al., 2015):

(1) Pressure initiates contractions by altering intrinsic Ca$^{2+}$ dynamics in LMCs.
(2) Imposed forward flow weakens or abolishes contractions.
(3) Outflow pressure elevation strengthens contractions.
(4) NO extends the range of pressures over which contractions propel lymph.
(5) Contractions propagate backwards (relative to the direction of lymph flow) in presence of NO and forwards in its absence.

Because lymphatic pressures and flows are poorly characterized and cannot easily be manipulated *in vivo*, we set out to test these predictions using cannulated, pressurized *ex vivo* preparations of rat mesenteric lymphatic vessels. The concepts underlying prediction 1 are not addressed here as they are firmly established by previous electrophysiological studies, including several using rat mesenteric lymphatics (Davis & Zawieja, 2025; von der Weid et al., 1996, 2014; Zawieja, Castorena-Gonzalez, To et al., 2018; Zawieja et al., 2019). Predictions 2 and 3 have been tested previously (Davis et al., 2012; Gashev et al., 2002; Scallan et al., 2012) but could be re-examined under the specific conditions used in the present study.

Prediction 4 is contradicted to some degree by a study in mice (Scallan & Davis, 2013) but could benefit from being tested over a wider pressure range and in a different network and species. Prediction 5 is the key hypothesis derived from the model (Kunert et al., 2015) that addresses the competing effects of flow and pressure, but it has never been tested experimentally. As the contractile behaviour of collecting lymphatic vessels varies widely between species and region, aggregate testing of these hypotheses under uniform experimental conditions would be optimal. Ideally experiments would be performed on vessels from mice so that predictions 4−5 could be tested with and without genetic deletion of NO. However murine lymphatic collectors from the abdominal and thoracic cavities lack propulsive lymphatic contractions (Zawieja, Castorena-Gonzalez, Scallan et al., 2018), and we have been unable to demonstrate flow-induced inhibition of contractions in vessels taken from multiple regions of the mouse (Scallan & Davis, 2013; Zawieja, Castorena-Gonzalez, Scallan et al., 2018) – possibly because their small size (50−80 μm, i.d.) dictates the use of relatively high-resistance cannulae across which substantial pressure drops limit the flow rate.

For these reasons we chose the mesenteric collecting vessel of the rat as our experimental model to test the individual and combined effects of flow and pressure on phasic lymphatic contractions. Unbranched, multi-valve segments could be easily located and cannulated, and internal vessel diameters in the range of 160−180 μm allowed the use of lower-resistance cannulae. Although less sensitive than the TD to flow, the SS changes associated with phasic contractions of rat mesenteric collectors are known to produce NO from the valve sinus regions (Bohlen et al., 2009, 2011), and in some cases contractions are substantially inhibited by imposed flow (Gashev et al., 2002; Scallan et al., 2012). With our system, independent control of pressure at either vessel end enabled testing of responses to imposed flow (with minimal midpoint pressure change), elevated inflow pressure (with consequent elevated flow) or elevated outflow pressure (in the absence of imposed flow). In each scenario the frequency, amplitude, conduction direction and conduction distance of the phasic contraction waves were compared in the presence or absence of NO production. The results help clarify our understanding of how the active lymph pump is regulated by the mechano-biological forces of pressure and flow.

## Methods

### Animal protocols

Male Sprague–Dawley rats were purchased from Envigo-Harlan (Indianapolis, IN, USA). All procedures were reviewed and approved by the University of Missouri Animal Care and Use Committee (#9797) and complied with the standards stated in the 'Guide for the Care and Use of Laboratory Animals' (National Institutes of Health, revised 2011). The animals had free access to food and water. Both male and female rats were used, but no attempt was made to analyse the results by sex.

### Vessel isolation

After the animals (130–220 g body weight) were anaesthetized with ketamine/xylazine (100/10 mg/kg, I.P.), an abdominal midline incision was made, and the entire small intestine was removed, pinned out in a 60 mm Sylgard-coated dish and covered with Krebs solution. Unbranched segments of collecting lymphatic vessels were isolated from the duodenal region. Immediately after their isolation vessel segments were transferred to a dissection chamber containing Krebs-albumin solution for removal of fat and connective tissue while pinned with short pieces of 40 μm wire. The animal was then killed with an overdose of ketamine/xylazine followed by cervical dislocation.

### Cannulation, pressure control and diameter tracking

Single-vessel segments containing 3−4 valves (i.e., 2−3 complete lymphangions) were cannulated at each end with a glass micropipette mounted on a Burg-style V-track system (Davis et al., 2012) and transferred to the stage of a Zeiss Axiovert 200 microscope. Both micropipettes were approximately 100 μm (outer diameter) at their tips, and the same two micropipettes were used for all experiments. After both the micropipettes were pressurized briefly to 10 cmH$_2$O from a standing reservoir, the distance between the two pipettes was adjusted to remove slack from the vessel; otherwise axial buckling at higher-pressure levels led to inaccurate diameter tracking. After this the pressure was reduced to and maintained at 3 cmH$_2$O for an equilibration period lasting from 45 to 60 min at 37°C. For the entire experiment a peristaltic pump exchanged the bath solution continuously with fresh Krebs solution at a rate of 0.5 ml/min. To enable computer control of pipette pressures the pipette connections were switched from the standing reservoir to a microfluidic pressure control system (OB1; Elveflow, Paris) by which input ($P_{in}$) and output ($P_{out}$) pressures could be independently controlled using a custom LabVIEW program (National Instruments Corp, Austin, TX, USA). The microscope image was digitized at a magnification of 0.98 pixels/μm using a Basler camera (model a2A1920-160 um; Ahrensburg, Germany), in synchrony with pressure and diameter data, and recorded in AVI format (Davis et al., 2011). Pressure control, acquisition and diameter tracking programmes are available online (Davis, 2023a, b).

## Contractile function measurements

Internal diameter was tracked continuously throughout the experiment (Davis, 2005). Afterward custom LabVIEW programs detected end diastolic diameter (EDD), end systolic diameter (ESD) and contraction frequency (FREQ) on a contraction-by-contraction basis. Each parameter was averaged over a 2−5 min period and used to calculate the following indices of lymphatic contractile function:

$$\text{Contraction amplitude (AMP)} = \text{EDD} - \text{ESD} \quad (1)$$

$$\text{Normalized contraction amplitude} = \left(\frac{\text{AMP}}{D_{\text{MAX}}}\right) \times 100 \quad (2)$$

$$\text{Ejection fraction (EF)} = \left[\frac{\text{EDD}^2 - \text{ESD}^2}{\text{EDD}^2}\right] \quad (3)$$

$$\text{Fractional pump flow (FPF)} = \text{EF} \times \text{FREQ} \quad (4)$$

where $D_{\text{MAX}}$ represents the maximum passive diameter (obtained after incubation with calcium-free Krebs solution) at a given level of intraluminal pressure.

**Solutions.** All chemicals were obtained from Sigma (St. Louis, MO, USA), excluding bovine serum albumin (BSA), which was obtained from US Biochemicals (catalogue no. 10856; Cleveland, OH, USA). Krebs solution contained (in mM): 146.9 mM NaCl, 4.7 mM KCl, 2 mM CaCl$_2$·2H$_2$O, 1.2 mM MgSO$_4$, 1.2 mM NaH$_2$PO$_4$·H$_2$O, 3 mM NaHCO$_3$, 1.5 mM Na-HEPES and 5 mM D-glucose (pH 7.4). The Krebs-albumin solution was identical except for the addition of 0.5 g/100 ml purified BSA. During the cleaning and temperature equilibration steps the bath and pipettes both contained Krebs-albumin, after which the external bath was changed to Krebs solution. Ca$^{2+}$-free Krebs solution, identical to Krebs solution, except that 3 mM EDTA was substituted for CaCl$_2$, was used to obtain passive diameters at every relevant pressure at the end of the experiment.

## Servo-null pressure measurements

To verify the extent to which the cannulating pipettes were matched for the imposed flow protocols an unbranched segment of vessel was cannulated, and a small hole was made using a pilot pipette as close as possible to the vessel midpoint. The pilot pipette was then replaced with a servo-null micropipette (tip diameter ≈5 μm) to measure midpoint luminal pressure, while $P_{\text{in}}$ and $P_{\text{out}}$ were changed according to protocol 1. A servo-null micro-pressure system (model 4, Instrumentation for Physiology & Medicine, San Diego, CA, USA) was used to make the pressure ($P_{\text{sn}}$) measurements. Before the protocol

1 pressure steps, the $P_{\text{sn}}$ calibration was checked and adjusted (if needed) while stepping $P_{\text{in}}$ and $P_{\text{out}}$ together between 1 and 5 cmH$_2$O; the calibration was re-checked after the test.

## Enhancement of NO bioavailability

To ensure that the vessels were capable of producing NO-mediated responses, the Krebs-albumin solution introduced into the vessel lumen was supplemented with L-arginine (1 mM; 13 vessels) and in some cases also included N$^{\text{(omega)}}$hydroxy-nor-L-arginine (100 μM; 2 vessels) (Sigma-Aldrich) to enhance the production and inhibit the breakdown, respectively, of NO produced either from oscillatory SS occurring during spontaneous contractions or steady SS during imposed flow.

## Pipette resistance matching

To minimize midpoint pressure changes during protocol 1 a microforge was used to match the resistances of the inflow and outflow pipettes by fire-polishing the pipette tips to similar internal diameters. Resistance matching was tested in two separate experiments in which a non-valved lymphatic segment was cannulated, and pressure at the vessel midpoint was measured using a servo-null micro-pipette while performing the pressure steps used in protocol 1. An example of this test is shown in Fig. 1*A*. Except for brief $P_{\text{sn}}$ transients when $P_{\text{in}}$ was changed slightly before $P_{\text{out}}$, $P_{\text{sn}}$ increased by < 0.1 cmH$_2$O during equal-but-opposite steps in $P_{\text{in}}$ and $P_{\text{out}}$.

## Protocols

The following protocols were performed on each vessel, beginning with a 2-min control period and ending with a 2-min recovery period.

(1) **Imposed flow**. Starting from equal pressures of 3 cmH$_2$O $P_{\text{in}}$ was elevated and $P_{\text{out}}$ lowered by the same amount, in two steps of 1 cmH$_2$O, each lasting for 2 min.
(2) **Elevated outflow pressure**. Starting from $P_{\text{in}} = P_{\text{out}} = 1$ cmH$_2$O, $P_{\text{out}}$ was elevated in 2 cmH$_2$O steps to 5 cmH$_2$O, each lasting for 2 min.
(3) **Elevated inflow pressure** (imposed flow + increased inflow pressure). Starting from $P_{\text{in}} = P_{\text{out}} = 1$ cmH$_2$O, $P_{\text{in}}$ was elevated in 2 cmH$_2$O steps to 5 cmH$_2$O, with each step lasting for 2 min.

The time interval between protocols varied. Typically 2 min was sufficient for recovery of a normal phasic contraction pattern, but about half the vessels needed longer recovery (up to 10 min) from protocol 1. However,

in all protocols vessels used for data analysis recovered a contraction amplitude $\geq$75% of control.

After completion of protocols 1$-$3 for most vessels L-NAME (100 µM) was administered in the bath solution for 20$-$30 min, and protocols 1$-$3 were repeated. In preliminary experiments protocols 2$-$3 used a larger pressure range with more steps, but this resulted in longer experiments in which vessels tended to fatigue. Those protocols were subsequently shortened to the ones listed above; however when appropriate, data from the longer protocols (using only data from the same $\Delta$P steps) were included in the data analysis.

(4) **Pressure range for spontaneous contractions**. In separate protocols the influence of SS-induced NO production on the pressure range over which phasic contractions occurred (prediction 4) was determined. Each vessel was first equilibrated at 3 cmH$_2$O and then subjected to protocol 1 to assess its responsiveness to imposed flow. Vessels with 1 mM L-arginine in the lumen responded to flow (Fig. 1*C*), whereas those exposed to L-NAME but without L-arginine supplementation did not (Fig. 1*B*). After a normal contraction pattern returned at 3 cmH$_2$O, $P_{in}$ and $P_{out}$ were simultaneously lowered to 2, 1, 0.5, 0.2 and 0.1 cmH$_2$O for 1$-$5 min at each pressure (depending on the contraction frequency), while diameter was recorded near the vessel midpoint. Subsequently pressures were returned to 3 cmH$_2$O for 5 min and then increased to 5, 8, 10 and 15 cmH$_2$O for 1 min at each pressure, while diameter was recorded. In two vessels pressure was raised further to 20 and 30 cmH$_2$O, pressures at which no further increases in frequency were noted and contraction amplitudes became negligible;

however these pressures appeared to damage the vessels irreversibly and so 15 cmH$_2$O was the pressure limit used in subsequent experiments. The average amplitude, frequency and FPF at each pressure were determined off-line, and the data were segregated into control (flow-responsive) and L-NAME treated vessels.

## Determination of passive diameter

At the end of each experiment the bath was exchanged with Ca$^{2+}$-free Krebs for 20 min followed by combined $P_{in}$ + $P_{out}$ pressure steps from 0.1 to 15 cmH$_2$O to generate a passive pressure-diameter curve.

## Data analysis

The data from each protocol consisted of a video file in JPEG-compressed AVI format and a time-series file detailing the time, pressures and tracked vessel diameter at a location near the vessel midpoint for each video frame. Protocols 1–3 each consisted of four stages (control, steps 1 and 2, recovery), each nominally 2 min long at video frame rates between 29 /s and 74 /s and each consisting of 1624 $\times$ 200 pixels or 1920 $\times$ 250 pixels covering a field of view 3.82 $\times$ 0.47 mm or 4.42 $\times$ 0.59 mm, respectively.

## Determination of conduction wave direction

Depending on the spontaneous contraction frequency, between 1 and 40 contractions were analysed during each stage. The file sections corresponding to each stage were isolated, and the video segment was processed

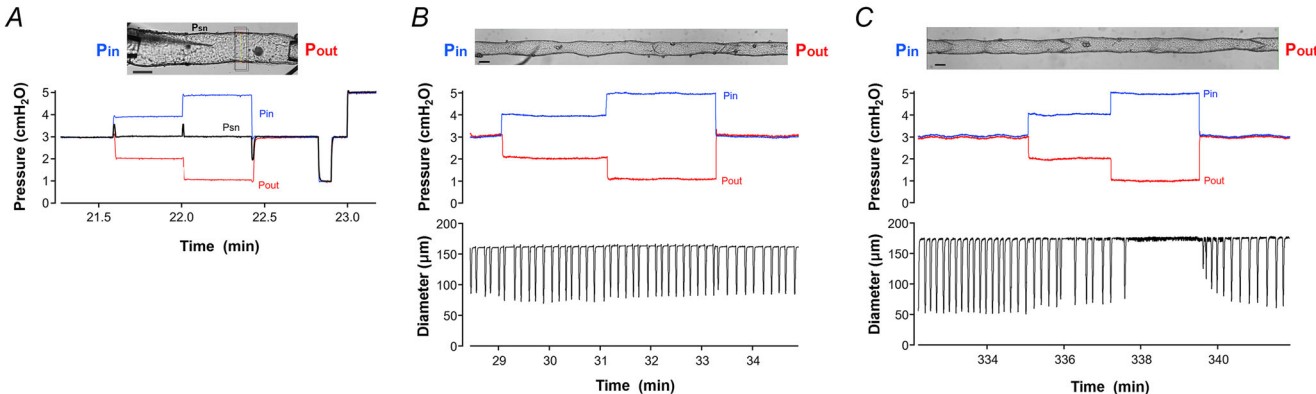

**Figure 1. Responses to imposed flow**
*A*, test of balanced pipette resistances using a passive, non-valved segment. The distances from the tips of the $P_{in}$ pipette to the $P_{sn}$ pipette and from the $P_{out}$ pipette to the $P_{sn}$ pipette were 680 and 693 µm, respectively. Steps at the end verify the accuracy of the $P_{sn}$ pipette calibration. *B*, flow test on an unresponsive vessel that showed almost no change in either contraction amplitude (AMP) or contraction frequency (FREQ) during imposed flow. *C*, flow test on a vessel with L-arginine in the luminal solution. During the second step FREQ and AMP decreased prior to complete cessation of contractions followed by gradual recovery of both when imposed flow ended. Scale bars = 100 µm.

using the Python program LySS_contraSTM_Spectro (see Acknowledgements) to obtain a spatiotemporal map (STM) of the contractions (Fig. 2*A*). The output of this program was an image corresponding to the vessel diameter at every frame time and every axial location along the vessel. The conduction direction of each contraction wave was determined using a custom LabVIEW program that detected the pixels at the leading edge of each contraction band in the STM and fitted them to a first-order polynomial to determine the slope (Fig. 2*B*). Nearly all contractions initiated at or near one of the vessels ends, with each solid band reflecting a contraction that conducted over the entire length of the vessel and with the intensity of the band inversely proportional to the strength of the contraction (i.e. darker bands = stronger contractions; Fig. 2*A*). Antegrade (forward) contractions, in the direction of normal lymph flow, had a positive slope ($\Delta$distance/$\Delta$time) with a point of origin near the inflow cannula. Retrograde (backward) contractions had a negative slope with a point of origin near the outflow cannula. The former were assigned a value of $+1$; the latter were assigned a value of $-1$. Rare contractions originating in the middle of the segment were assigned a fractional value between $-1$ and 1 in proportion to their site of origin. The numbers representing conduction direction for each set of contra-

ctions during each 2-min sequence were averaged to obtain a single value for use in subsequent statistical analyses. For example, a vessel initially exhibiting 23 contractions in the retrograde direction and 2 in the antegrade direction would be assigned a conduction direction value of $-0.84$ during the control period.

## Determination of contraction wave conduction distance

Once an action potential is generated from a pacemaking site, an electrical activation begins to conduct along the vessel, initiating a contraction at each point. The resulting wave of contraction that we observe normally conducts over the entire length of the vessel. For this analysis the predetermined thresholding feature of the STM analysis program was used to avoid subjective assessments of contraction wave strength. Under some conditions the observed wave may extinguish and then may or may not reappear in the adjacent region; these phenomena (often occurring during the second step of a protocol) are evident by shortened or interrupted bands, respectively, in the STM. Because of the fixed threshold, the wave can seem to be interrupted when it may simply be locally weakened below threshold. To assess the influences of pressure and flow on the extent of contraction wave conduction the

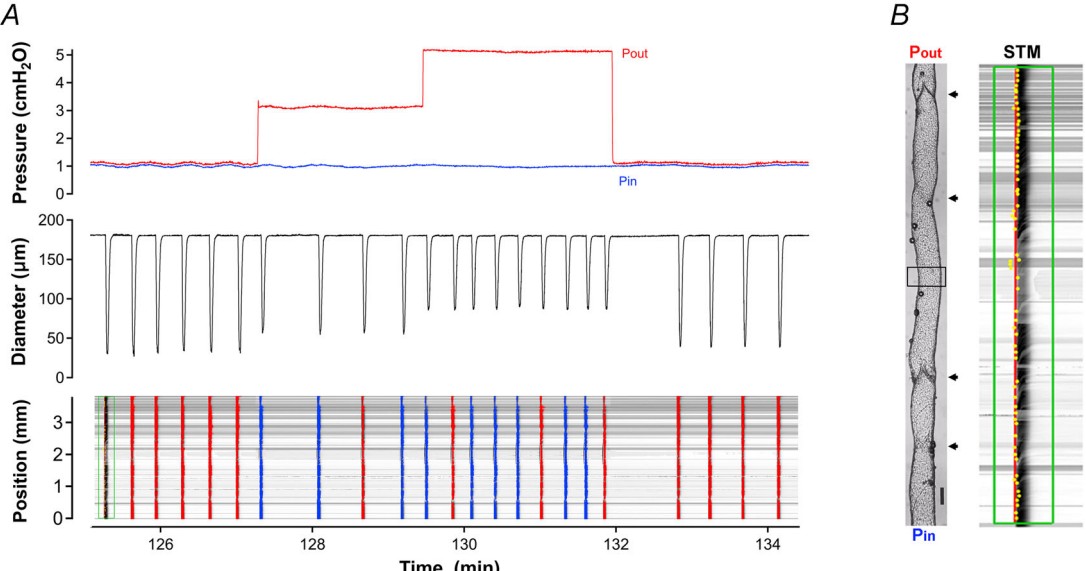

**Figure 2. Determination of conduction wave direction**
*A*, example recording from a vessel subjected to protocol 2 ($P_{out}$ steps) with the corresponding spatiotemporal map (STM) shown at the bottom. The faint horizontal lines in the STM are tracking artefacts associated with residual fat and connective tissue. *B*, the vessel image (at left) is aligned with the STM; arrows indicate valve locations. The LabVIEW-generated ROI for edge detection is overlaid (green box at right) on the first contraction band of the STM, showing detected edges (yellow dots) and the line of best fit to those points (in red). Based on the slope analysis for each contraction antegrade contraction waves in *A* are coloured in blue (conduction away from $P_{in}$ pipette) and retrograde waves in red (conduction away from $P_{out}$ pipette). Conduction was predominantly in the retrograde direction initially, switched to a predominantly antegrade direction when $P_{out}$ was elevated and returned to the retrograde direction after $P_{out}$ was restored to the control level.

relative distance travelled by each wave was measured in each STM for the respective control, step 1 and step 2 intervals, and recovery period. The relative distance was defined as that over which supra-threshold contraction was detected, and the results were grouped in a similar manner as those for conduction direction.

### Analysis of contraction strength and frequency

For testing predictions 2−3, contraction AMP and FREQ for protocols 1−3 were measured from the original internal diameter records near the vessel midpoint and averaged over the respective stages (control, steps 1 and 2, recovery).

### Statistical analyses

The data were analysed using custom LabVIEW programs and compiled in Excel. Raw data plots were made in IGOR (Wavemetrics, Oswego, OR, USA). Graphs and statistical tests were performed in Prism (v10; GraphPad, San Diego, CA, USA) and plotted as mean ± SD. Two-way ANOVAs (with repeated measures or mixed models) and Sidak's *post hoc* tests were used to compare contraction parameters (amplitude, frequency, FPF) between stages of protocols 1–3 and across groups (control *vs.* L-NAME-treatment). One-way ANOVAs with Tukey's *post hoc* tests were used for analyses of conduction direction and distance. For protocol 4 two-way ANOVAs (with repeated measures or mixed models) and Sidak's *post hoc* tests were used to compare contraction parameters at each pressure level between groups (control *vs.* L-NAME treatment). Vessels without any contractions during a particular recording segment were assigned a frequency value of zero without a corresponding value for amplitude; this resulted in missing and lower *n*-values for amplitude analyses, dictating the use of mixed rather than two-way ANOVAs in some cases and with insufficient numbers for some statistical tests. Specific statistical tests, vessel numbers (*n*) and animal numbers (*N*) are shown or stated in the figures or figure legends.

### Results

Rat mesenteric lymphatics containing 2−3 complete lymphangions generally showed inconsistent responses to imposed flow *ex vivo*. Our initial experiments on several different batches of rats from two different vendors produced neither consistent nor robust responses to imposed flow using protocol 1 (1 out of 13 vessels); see example in Fig. 1*B*. To rule out the possibility of impaired LEC viability we performed a separate series of experiments to test whether the endothelium-dependent vasodilator ACh could evoke NO production and an associated change in contraction. ACh (1 or 10 μM) was applied to the bath after a vessel had established a regular contraction pattern. Eleven out of 11 vessels responded to either or both concentrations with a decrease in contraction frequency to 69% ± 8% of control for 1 μM, $p = 0.0058$, and to 72% ± 11% of control for 10 μM, $p < 0.0132$ ($n = 11$ and 10, respectively) by one-way ANOVA. The responses to ACh were completely blocked by 20 min exposure to L-NAME (100 μM). We therefore suspected that the relatively long vessel segments used for our protocols were limiting the absolute flow/SS levels that could be imposed in protocol 1. For this reason we performed an additional series of tests to assess the responses of two-valve mesenteric lymphatic segments to protocol 1. In those experiments five out of six vessels responded to imposed flow (at $P_{in} = 5$ cmH$_2$O, $P_{out} = 1$ cmH$_2$O, contraction frequency decreased to 54% ± 33% of control, $p = 0.0484$; $n = 6$; by one-way ANOVA), and that response was completely blocked by L-NAME (100 μM). Thus the limited response of 3−4-valve segments to imposed flow likely reflected higher resistances of those segments rather than impaired NO signalling or bioavailability.

To ensure that flow/SS mechanisms were operative and to be able to test the predictions of the Kunert et al. (2015) model, we borrowed methods used in studies of arteries and cultured endothelial cells to enhance NO bioavailability and/or minimize its degradation (Durante, 2022; Kuo et al., 1992). Supplementation of the Krebs-albumin solution filling the pipettes connecting tubing and vessel lumen with 1 mM L-arginine resulted in substantial or sometimes complete cessation of phasic contractions in most (16 out of 19) vessels during protocol 1 (imposed flow; see example in Fig. 1*C*) and/or protocol 3 (inflow pressure elevation). Indeed flow-induced inhibition was often so strong that it could not be blocked subsequently by L-NAME (which competes for the L-arginine binding site on endothelial NO synthase) at the concentration (100 μM) typically used in other lymphatic and artery studies. A 10-fold higher concentration of L-NAME (1 mM) was more effective at inhibiting flow-induced responses but led to increasingly smaller contraction amplitudes over time in the absence of flow or pressure changes, suggesting additional off-target effect(s). For these reasons we compared the responses of 3−4-valve vessels treated with L-arginine to enhance NO bioavailability to the responses of similar vessels without L-arginine treatment but with 100 μM L-NAME present in the bath to inhibit any production of NO that might otherwise occur.

The datasets for summary analyses were partitioned into vessels with predominantly antegrade conduction direction and vessels with predominantly retrograde contraction direction. Combining these groups would

have obscured the effects of pressure/flow on determining the conduction direction. A few vessels had an approximately equal mixture of conduction directions (5 out of 60 for protocol 1; 0 out of 35 for protocol 2; 0 out of 40 for protocol 3) in the control periods and thus were excluded from the summary analysis of conduction direction.

### Imposed flow without a midpoint pressure change (protocol 1)

Imposed flow caused a significant slowing of phasic contractions and significant reductions in both normalized contraction amplitude and FPF. The data are summarized in Fig. 3*A* (open circles). Vessels treated with L-NAME had significantly higher phasic frequencies and tended to have lower amplitudes, as reported in previous studies (Scallan & Davis, 2013). The contraction parameters of vessels treated with L-NAME were resistant to the inhibitory effects of imposed flow (Fig. 3*A*, closed circles).

Next, STMs were analysed for the effect of imposed flow on the direction of contraction wave conduction. At the start of protocol 1 the dominant conduction direction was retrograde (38 retrograde, 17 antegrade, 5 mixed from a total of 60 imposed flow tests) – a trend (63% retrograde) consistent with that of *ex vivo* vessels in a previous study (Castorena-Gonzalez et al., 2020). The data were segregated by (1) the predominant direction of conduction prior to flow steps and (2) whether or not the vessel was responsive to imposed flow (i.e. exhibited flow-induced inhibition of amplitude and/or frequency). A few vessels with predominantly antegrade conduction initially switched to retrograde conduction during imposed flow, but this trend was not significant (Fig. 3*B*). Vessels with predominantly retrograde conduction had a stronger tendency to switch to antegrade conduction during imposed flow, but that response also was not statistically significant.

Imposed flow produced a more consistent effect on the distance over which supra-threshold contraction waves conducted. In flow-responsive vessels both flow steps prevented many contraction waves from conducting over the entire length of the vessel (Fig. 3*C*), and the velocities of those waves were also reduced (not shown). This effect was largely, but not completely, blocked by L-NAME (compare left and right panels of Fig. 3*C*). Examples of vessels showing these responses to imposed flow are shown in Fig. 3*D,E*. The inhibitory effects of imposed flow often lasted into the recovery period (as evident in Fig. 3*D,E*), a pattern previously observed in arteries that can be explained by the washout of SS-induced NO production abruptly ceasing upon termination of the flow step (Kuo

et al., 1990). Collectively these results show that imposed flow inhibited contraction amplitude, frequency and FPF, impaired the complete conduction of supra-threshold contraction waves and tended to reverse the direction of contraction wave conduction, particularly in vessels showing initial retrograde conduction.

### Elevated outflow pressure (protocol 2)

Flow-responsive and unresponsive vessels with 3−4 valves responded similarly to $P_{out}$ elevation, with normalized amplitude decreasing significantly and normalized frequency increasing significantly (Fig. 4*A*). FPF changes were intermediate. As in the previous protocol L-NAME treatment resulted in a significant elevation in basal frequency and FPF (as signified by *p*-values shown above the open and closed symbols in Fig. 4*A*). These results suggest that basal production of NO due to pulsatile SS lowers the contraction frequency significantly but does not alter the amplitude, frequency or FPF responses to $P_{out}$ elevation.

At the start of protocol 2 the dominant direction of conduction was again retrograde (27 out of 35 tests; 77% retrograde). Of vessels with initial antegrade conduction only two were responsive to flow, and their conduction direction did not change with $P_{out}$ elevation (Fig. 4*B*, upper left panel). In presence of L-NAME to inhibit SS-induced NO production four out of six flow-unresponsive vessels with initial antegrade conduction switched to retrograde conduction when $P_{out}$ was raised, but the overall response was not significant (Fig. 4*B*, lower left panel). Vessels with predominantly retrograde conduction did not show a significant switch in conduction direction during $P_{out}$ elevation regardless of whether they were flow-responsive (Fig. 4*B*, right panels).

$P_{out}$ elevation tended to reduce the distance travelled by the contraction wave and that effect became significant for L-NAME-treated vessels (Fig. 4*C*). Examples of conduction direction switching from antegrade to retrograde with $P_{out}$ elevation are shown in Fig. 4*D,E*, with the vessel in Fig. 4*E* also showing some impairment in overall conduction distance by the contraction wave weakening to below threshold at three points.

For two vessels with initial antegrade conduction the responses to smaller $P_{out}$ steps were tested (Fig. 5) and revealed that step increases in $P_{out}$ as small as 0.2 cmH$_2$O could induce (although sometimes only transiently) a switch from antegrade to retrograde conduction direction; the direction again reversed when $P_{out}$ was lowered. The results in Figs 4 and 5 suggest that the elevation of $P_{out}$ tends to maintain or promote retrograde conduction (Fig. 4*B*) with a slight inhibitory effect on conduction distance (Fig. 4*C*).

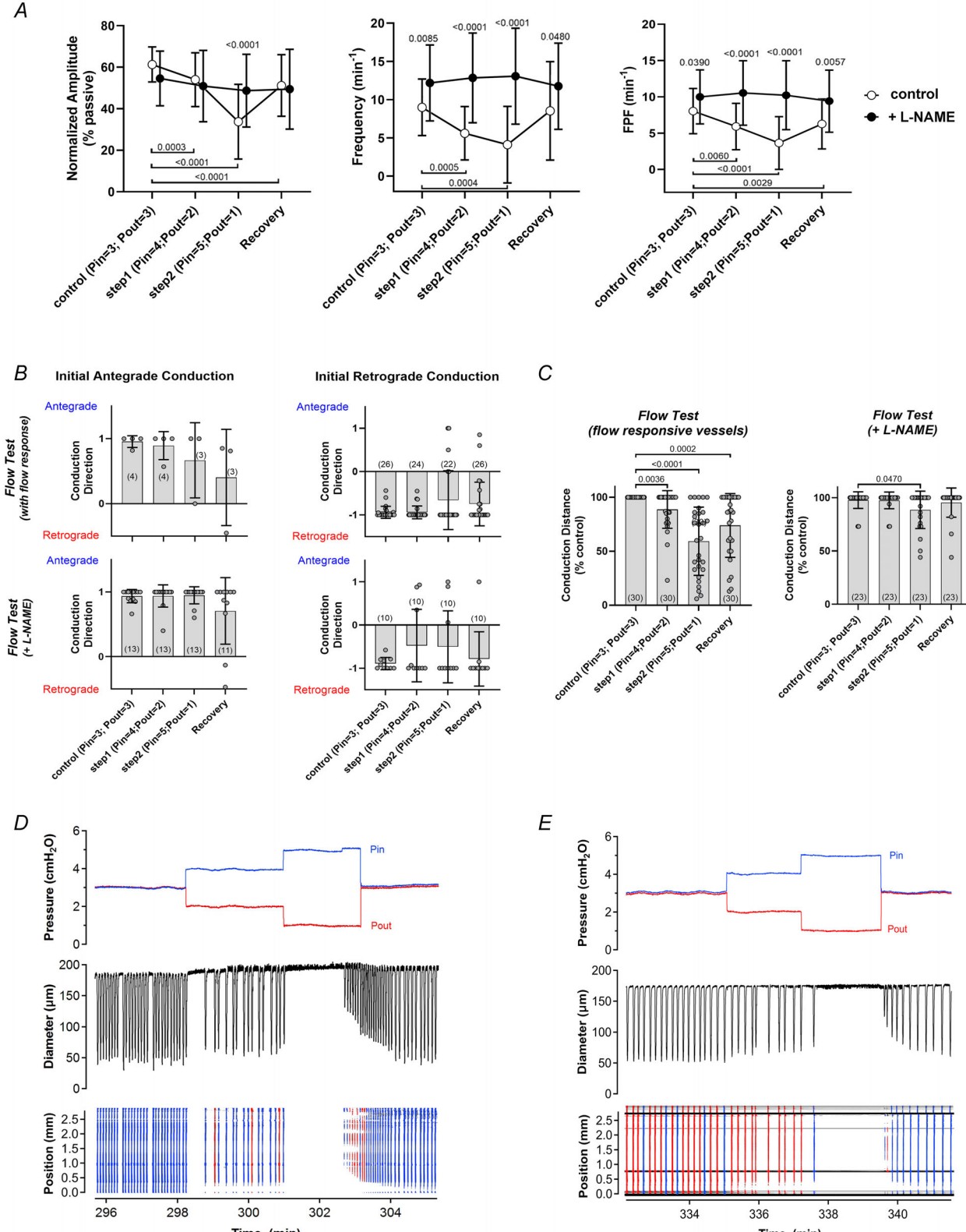

**Figure 3. Protocol 1: Imposed flow**

*A*, summary of changes in normalized amplitude, frequency and fractional pump flow (FPF) in response to imposed flow (protocol 1). Lower bars with *p*-values show comparisons between responses during imposed flow and

the respective initial parameters prior to flow (in absence of L-NAME). No comparisons among groups with L-NAME treatment were significant. The *p*-values without bars indicate significance levels between groups with and without L-NAME treatment (two-way ANOVA with Tukey's *post hoc* tests). Other comparisons (unmarked) were not significant at $p < 0.05$. Control: $N = 12$, $n = 29-30$; L-NAME: $N = 12$, $n = 26-27$. *B*, summary of changes in conduction direction during imposed flow (protocol 1), with vessels segregated based on their predominant conduction direction (antegrade $= +1$; retrograde $= -1$) before the start of the protocol. One-way ANOVAs with Dunnett's *post hoc* tests were used for statistical testing. *C*, summary of changes in contraction distance (normalized to control) for the same groups of vessels. One-way ANOVAs with Dunnett's *post hoc* tests were used for statistical testing. *D*, example of a vessel with antegrade conduction that continued primarily antegrade conduction during imposed flow. Most waves during the flow steps failed to conduct supra-threshold contractions over the entire vessel length. *E*, example of a vessel with predominantly retrograde conduction that switched to antegrade conduction at the highest level of imposed flow (but with only a single antegrade contraction before contractions ceased) and failed to conduct to the inflow end; conduction continued in the antegrade direction during the recovery period. In both examples the contraction waves fell below the threshold amplitude for detection over part of the vessel length during imposed flow, and the conduction directions in the recovery periods were often the opposite to those in the control periods. The outflow end is at the top of each spatiotemporal map (STM). Antegrade waves are coloured blue; retrograde waves are coloured red.

### Elevated inflow pressure (protocol 3)

Selective elevation of $P_{in}$ produced a pressure increase across the entire length of the vessel, with simultaneous elevation in flow. In response to $P_{in}$ elevation the average contraction amplitude of vessels with NO signalling intact (Fig. 6*A*, open symbols) increased slightly during the first $P_{in}$ step but then decreased significantly during the second $P_{in}$ step when higher flow rates occurred (Fig. 6*A*, lower bars). Inhibition of NO (i.e. flow-unresponsive vessels treated with L-NAME) led to a decrease in the initial contraction amplitude, which then increased significantly during both $P_{in}$ steps (Fig. 6*A*, upper bars). With NO signalling intact frequency and FPF were not altered significantly by $P_{in}$ elevation, probably because the inhibitory effects of flow were offset by the stimulatory effects of elevated pressure. After L-NAME treatment frequency and FPF increased significantly during both $P_{in}$ steps (Fig. 6*A*, upper bars).

At the start of protocol 3 the majority of vessels exhibited retrograde contraction waves (27 out of 40 tests; 68% retrograde). The vessels with predominantly antegrade conduction did not significantly switch direction during $P_{in}$ steps (Fig. 6*B*, left panels), but vessels with predominantly retrograde conduction showed a trend to switch to antegrade conduction during each of the $P_{in}$ steps (Fig. 6*B*, upper right panel, with example shown in Fig. 6*E*). This switch was more pronounced (and significant) in flow-unresponsive vessels treated with L-NAME (Fig. 6*B*, lower right panel). During $P_{in}$ steps supra-threshold contraction waves failed to conduct over the entire vessel length (a significant effect) regardless of whether NO signalling was intact (Fig. 6*C*, example shown in Fig. 6*D*). These results suggest that $P_{in}$ steps have mixed effects on amplitude and frequency, tend to promote antegrade conduction and inhibit the conduction of contraction waves that are everywhere supra-threshold.

### The effects of SS-induced NO production on the pressure range of spontaneous contractions (protocol 4)

The results of protocol 4 are presented in Fig. 7. Vessels that were responsive to flow did not have significantly wider pressure ranges over which spontaneous contractions developed, nor were the contractions significantly stronger (larger amplitudes) than those of vessels that were flow-unresponsive and treated with L-NAME. In fact L-NAME-treated vessels tended to have slightly *higher* frequencies and amplitudes at the lower pressures (see inserts in left and middle panels of Fig. 7), although those differences were not significant. The results reinforce the conclusion that NO production associated with pulsatile SS does not have a strong influence on the amplitude or frequency of spontaneous contractions.

### Discussion

A major goal of our study was to test predictions about the control of lymphatic pumping by mechano-biological forces made by the numerical model of Kunert et al. (2015) and, in a wider context, to understand how these forces interact to determine active lymph transport. Our results using multi-valve segments of rat mesenteric collectors with independent control of inflow and outflow pressures support the following principles regarding the control of lymphatic pacemaking and pumping. (1) Pressure elevation has a stimulatory effect on contraction amplitude over a limited pressure range and a stimulatory effect on contraction frequency over the entire physiological pressure range. (2) Imposed flow has inhibitory effects on both amplitude and frequency (Fig. 3*A*), whereas the effects of pulsatile flow/SS are not obvious, except to alter baseline contraction frequency and amplitude (Figs 3*A*, 4*A*, 5*A*). The latter effects

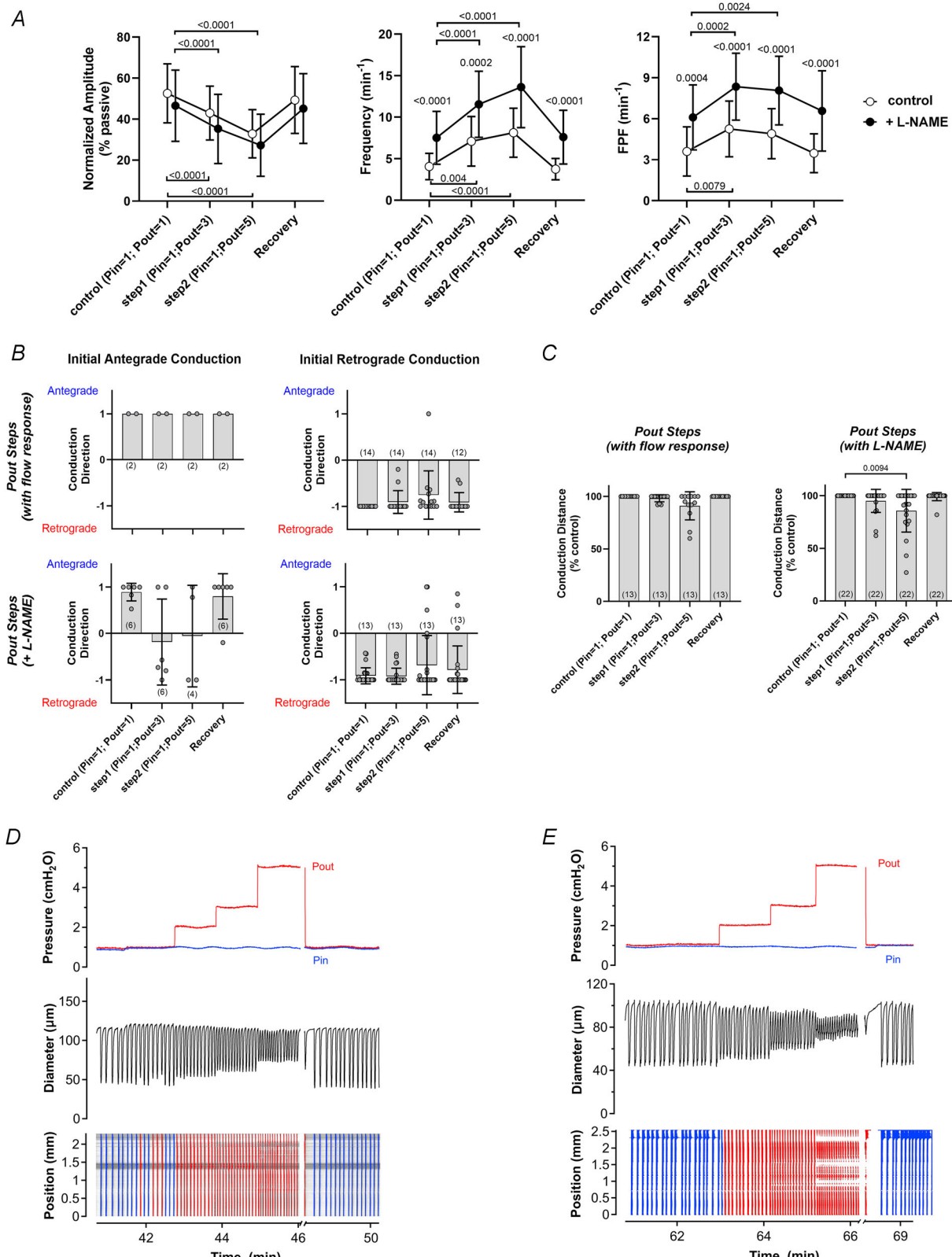

**Figure 4. Protocol 2: Selective elevation in outflow pressure**
*A*, summary of contraction amplitude (AMP), contraction frequency (FREQ) and fractional pump flow (FPF) changes during $P_{out}$ elevation (protocol 2). Lower bars with *p*-values show comparisons between the respective initial parameters and responses during $P_{out}$ steps in the absence of L-NAME (i.e. flow-responsive vessels). Upper bars with *p*-values show comparisons for flow-unresponsive vessels in presence of L-NAME. The *p*-values without bars

indicate significance levels between groups with and without L-NAME treatment. Unmarked comparisons were not significant at $p < 0.05$. Control: $N = 11$, $n = 18$; L-NAME: $N = 12$, $n = 24–27$. B, summary of changes in conduction wave direction during selective $P_{out}$ elevation. C, summary of changes in contraction wave conduction distance during selective $P_{out}$ elevation. D, example recording of a vessel with antegrade conduction that switched to retrograde conduction when $P_{out}$ was elevated. E, another vessel with a similar response to $P_{out}$ elevation but with a discontinuous contraction wave when $P_{out}$ was raised to 5 cmH$_2$O. The outflow end is at the top of each spatiotemporal map (STM). Antegrade waves are coloured blue; retrograde waves are coloured red. The same statistical tests as stated in the caption to Fig. 3 were used.

are consistent with a previous study in which NO produced by pulsatile SS subtly enhanced the rate of diastolic relaxation to reduce pacemaking frequency and secondarily increase contraction amplitude (Gasheva et al., 2006); however neither that result nor ours, both of which rely on comparisons between L-NAME treated and non-treated vessels, can exclude off-target effects of L-NAME, as the same L-NAME concentration used in both studies altered the baseline amplitude and frequency of lymphatic collectors from eNOS$^{−/−}$ mice (Scallan & Davis, 2013). The present study is the first to examine the effects of both flow and pressure on contraction wave conduction, with the following additional conclusions. (3) Contraction waves conduct predominantly in the retrograde direction when inflow and outflow pressures are equal. (4) Imposed flow has no significant effect on the direction of conduction but limits the distance that contraction waves conduct irrespective of their initial direction. (5) $P_{out}$ elevation does not change the direction of retrograde contraction waves but promotes the switching of antegrade waves to retrograde waves. (6) $P_{in}$ elevation (with consequent forward flow) does not alter antegrade wave direction but promotes the switching

of retrograde waves to antegrade waves. These effects are primarily or solely due to pressure, with the vessel segment experiencing the highest pressure dictating the pacemaking site. (7) $P_{in}$ elevation has equivalent effects on conduction direction irrespective of intact NO signalling, indicating that the associated forward flow does not override or significantly alter the effect of elevated pressure at the inflow end. Thus although imposed flow is inhibitory by suppressing contraction amplitude and frequency, and inhibiting the complete conduction of contraction waves, the influence of pulsatile SS is at best quite subtle compared to the influence of pressure. Finally (8) NO produced from pulsatile SS does not extend the pressure range over which spontaneous contractions are generated.

### Interpretation of our results in relation to the predictions of Kunert et al. (2015)

The model of Kunert et al. (2015) assumes that (1) pressure initiates contractions by altering the intrinsic Ca$^{2+}$ dynamics in LMCs. Mechanistically van Helden's studies (van Helden, 1993; von der Weid, 2019; von der Weid et al., 2008) pointed to the activation

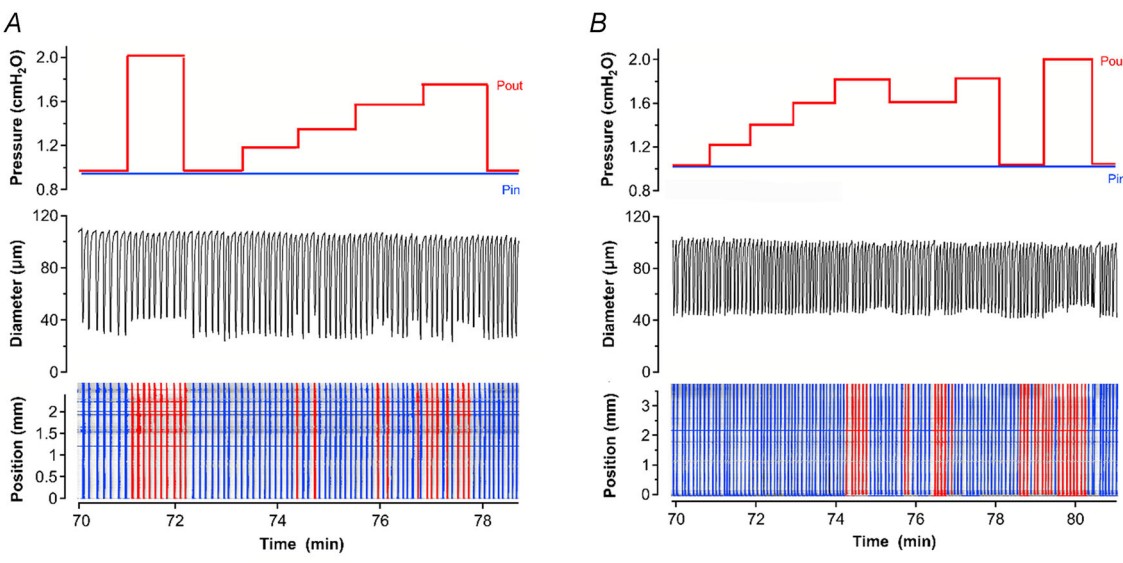

**Figure 5. Small elevations in outflow pressure can induce a change in conduction direction**
A, B, two examples of vessels initially showing predominantly antegrade conduction that switched to predominantly retrograde conduction in response to even small increments in $P_{out}$. Outflow end is at the top of each spatiotemporal map (STM). Antegrade waves are coloured blue; retrograde waves are coloured red.

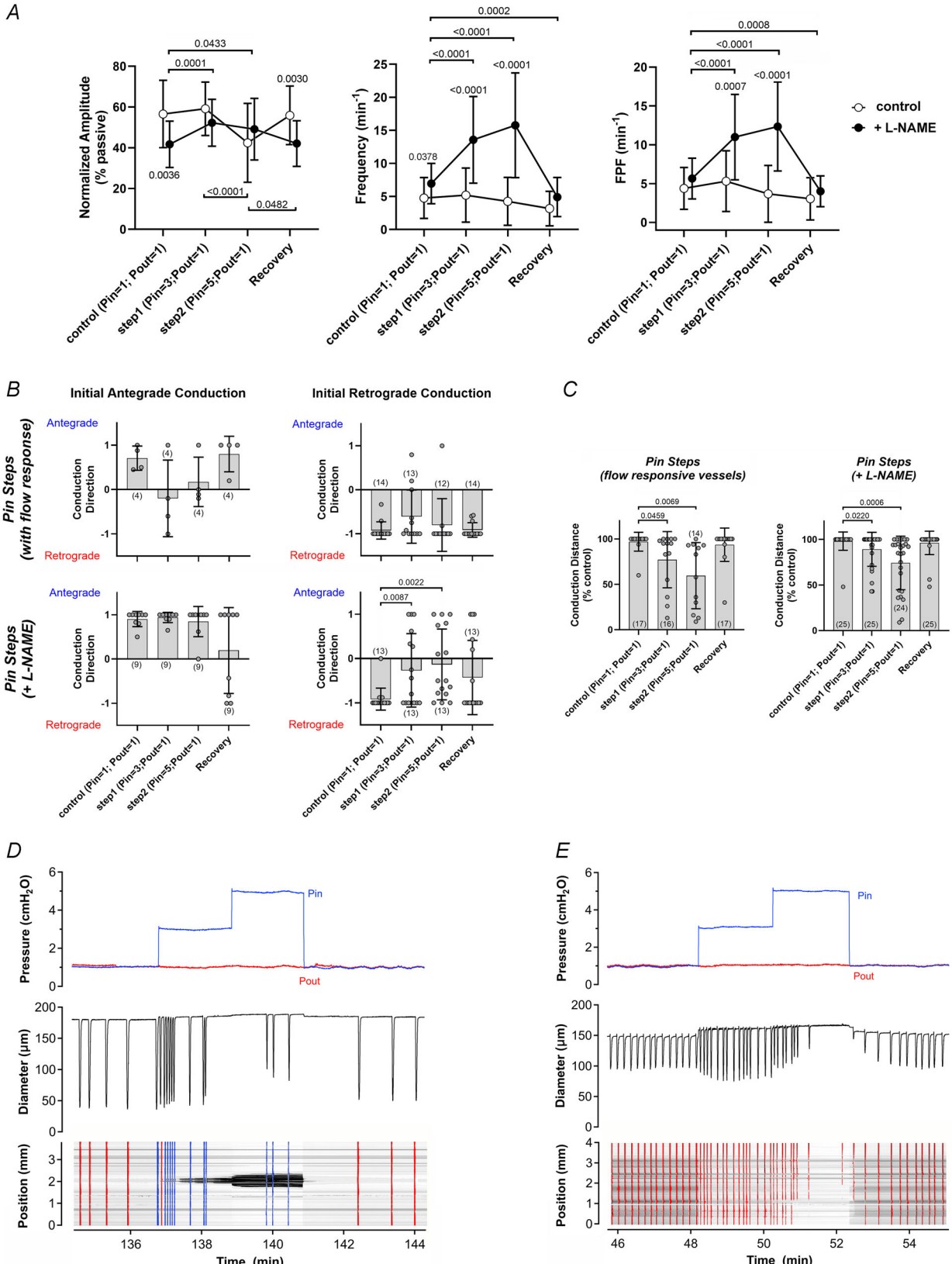

**Figure 6. Protocol 3: Selective elevation in inflow pressure**

*A*, summary of contraction amplitude (AMP), contraction frequency (FREQ) and fractional pump flow (FPF) changes during $P_{in}$ elevation (protocol 3). Lower bars with *p*-values show comparisons between the respective initial

parameters and responses to $P_{in}$ steps in the absence of L-NAME (i.e. flow-responsive vessels). Upper bars with *p*-values show comparisons for flow-unresponsive vessels in presence of L-NAME. The *p*-values without bars indicate significance levels between groups with and without L-NAME treatment. Unmarked comparisons were not significant at $p < 0.05$. Control: $N = 9$, $n = 17-24$; L-NAME: $N = 11$, $n = 21$. *B*, summary of changes in conduction wave direction during selective $P_{in}$ elevation. *C*, summary of changes in contraction wave conduction distance during selective $P_{in}$ elevation. *D*, example of a vessel with predominantly retrograde conduction that switched to antegrade conduction during $P_{in}$ elevation. The dark horizontal band is caused by a bubble that formed during the protocol on the outside of the vessel. *E*, example of a vessel with retrograde contractions that weakened and apparently ceased to conduct over the entire length of the vessel after $P_{in}$ was elevated. The outflow end is at the top of each spatiotemporal map (STM). Antegrade waves are coloured blue; retrograde waves are coloured red. The same statistical tests as stated in the caption to Fig. 3 were used.

of a calcium-activated $Cl^-$ conductance producing small ($1-2$ mV in amplitude) spontaneous, transient depolarizations (STDs) in LMCs, which summate to reach the threshold for triggering inward current through L- and/or T-type voltage-gated $Ca^{2+}$ channels, thereby generating a LMC action potential (Imtiaz et al., 2007; Lee et al., 2014). ANO1 was recently identified as the relevant plasmalemmal $Cl^-$ channel in murine LMCs (Zawieja et al., 2019), with $Ca^{2+}$ release from SR stores through $IP_3R1$ channels being the primary driver of ANO1 activity (Zawieja et al., 2023). Although neither the Kunert et al. (2015) model nor our present study advances our understanding of these mechanisms, our new findings reinforce the central contribution of pressure/stretch to the process (see below).

The Kunert et al. (2015) model predicts that (2) imposed forward flow weakens or abolishes contractions. This is confirmed by our results in Fig. 3*A*, although the concept had been previously established by earlier studies using similar *ex vivo* protocols for imposing flow (DuToit et al., 2024; Gashev et al., 2002, 2004, 2006, 2013). A new insight from our study is that steady flow inhibits the complete conduction of an otherwise fully entrained contraction wave (Figs 3*C* and 6*C*), although $P_{out}$ elevation also does this to a lesser extent (Fig. 4*C*).

The Kunert et al. (2015) model posits that (3) outflow pressure elevation 'increases contraction amplitude'. As shown in Fig. 3*B*, *D* and *E* of their paper, it is predicted that amplitude increases with pressure in the low-pressure range from 0 to 4 (units not specified) where frequency unphysiologically decreases, whereas at higher pressures contraction frequency moderately increases while contraction amplitude decreases. In the present study (Fig. 4*A*, left panel) and previous studies (Davis et al., 2012; Scallan et al., 2013) contraction amplitude *declines* as $P_{out}$ is moderately increased, in part because both frequency and tone are increased, which together compromise diastolic filling.

The Kunert et al. (2015) model predicts that (4) NO extends the range of pressures over which contractions propel lymph. The basis of this prediction is the apparent assumption that contraction-induced flow leads to an increase in NO concentration that, by inducing relaxation, is capable of triggering the next stretch-induced contraction: 'without NO production, phasic contractions are inhibited and can only occur at moderate transwall pressures, which are capable of providing the stretch-activated $Ca^{2+}$ spikes … with lower pressures, insufficient stretch activation occurs … at higher pressures, the stretch channels are always activated, resulting in stasis'. A subsequent model by the

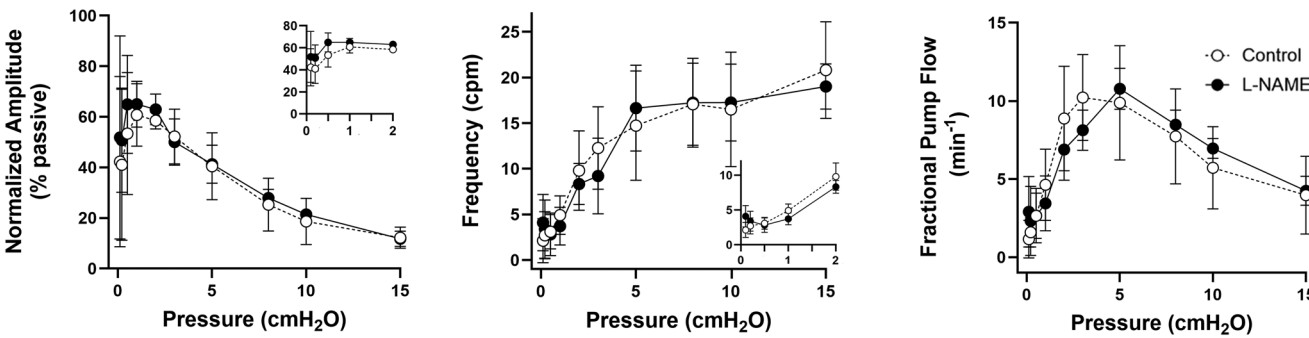

**Figure 7. Contraction parameters with and without intact NO signalling**
Contraction parameters as a function of pressure under control conditions (flow-responsive vessels) and after inhibition of NO signalling (L-NAME-treated vessels). No statistically significant differences in any parameter at any pressure were evident between the two groups. Inserts show expanded axes for contraction amplitude and frequency in the low-pressure range. Two-way (or mixed) ANOVAs with Tukey's *post hoc* tests were used for statistical tests. Control: $N = 5$, $n = 5$; L-NAME: $N = 6$, $n = 6$).

same group (Baish et al., 2016) extends this concept but also postulates that either stretch-mediated contractions or shear/NO-mediated cyclic oscillations in isolation can be triggered by external mechanical noise perturbations. To test hypotheses 4 and 5 of the Kunert et al. (2015) model it was important for us to verify that NO bioavailability was intact. The data in Figs 1 and 3 show that ʟ-arginine supplementation enhanced NO production to sufficient levels that forward flow imposed by relatively modest $\Delta P$ gradients substantially or completely inhibited spontaneous contractions in most (16 out of 19) vessels containing 3−4 valves. The results in Fig. 7 refute the predictions of Kunert et al. (2015) and Baish et al. (2016) in showing that no significant differences in contraction frequency occurred *at any pressure* in ʟ-arginine-treated vessels with intact flow responses compared to flow-unresponsive vessels treated with ʟ-NAME. Unlike previous studies we extended our observations to the low-pressure range (to 0.1 cmH$_2$O), where regular contractions persisted even at or beyond the point of vessel collapse, and occurred regardless of whether NO signalling was intact or inhibited (Fig. 7). In the high-pressure range contraction frequency reached a plateau (or sometimes even declined) while amplitude approached zero. It is possible that intact NO signalling might maintain contraction amplitude at higher pressures, but it was difficult to test this idea at pressures >15 cmH$_2$O, which appeared to damage the vessels. However from the right side of the amplitude graph in Fig. 7 no trend was evident to suggest that ʟ-arginine-treated vessels would have higher amplitudes (compared to ʟ-NAME-treated vessels) if higher pressures had been tested. Regardless, pressures >15 cmH$_2$O exceed the known physiological range of rodent lymphatic vessels (Hargens & Zweifach, 1977; Zweifach & Lipowsky, 1984) and thus are not relevant.

Finally, the Kunert et al. (2015) model predicts that (5) contractions propagate backwards in the presence of NO and forwards in its absence. Although multiple studies have documented both antegrade and retrograde conduction waves in lymphatic vessels (Benoit et al., 1989; Castorena-Gonzalez et al., 2018, 2020; Crowe et al., 1997; McHale & Meharg, 1992; Zawieja et al., 1993), only three have quantified the predominant conduction direction. Two *in vivo* studies reported 93% retrograde conduction for mouse popliteal collectors (Castorena-Gonzalez et al., 2020) and a 50:50 mixture for rat mesenteric collectors (Zawieja et al., 1993), but such measurements are potentially complicated by signals from converging branches. A study of guinea pig mesenteric vessels *in situ* reported 64% antegrade (orthograde) conduction (Crowe et al., 1997) in vessels cannulated and perfused/pressurized only from the inflow end. Measurements of contraction waves in *ex vivo* mouse popliteal vessels with controlled pressures at both ends found 68% retrograde conduction (Castorena-Gonzalez et al., 2020), which aligns with our data for rat mesenteric vessels (63%–77% retrograde waves in protocols 1–3). No studies prior to the present one have tested the effects of flow/NO on conduction direction. Our finding that retrograde conduction predominates both with NO signalling intact or blocked (Fig. 3*B*) refutes the prediction of Kunert et al. (2015) that pulsatile NO production promotes antegrade conduction. Our results were complicated by the necessity of segregating the data into vessels with initial antegrade conduction and those with initial retrograde conduction. However in both groups there was no clear pattern of a shift in conduction wave direction during the imposed flow protocol, nor was a significant shift evident when the two data sets were combined (not shown). In the selective pressure elevation protocols $P_{out}$ elevation (in which a pressure increase was limited to the last segment following closure of the outflow valve) tended to favour retrograde conduction (Fig. 4*B*), whereas $P_{in}$ elevation (with pressure and flow increasing throughout the vessel) tended to favour antegrade conduction (Fig. 6*B*, lower right panel); however the latter effect occurred irrespective of whether NO signalling was intact and only became significant when NO signalling was blocked. Collectively these results suggest that contraction waves are biased towards retrograde conduction regardless of NO production, and that the primary determinant of a shift in conduction direction is pressure. This conclusion is reinforced by the examples in Fig. 5 in which very slight changes in outflow pressure led to a switch from antegrade to retrograde conduction. Thus an emerging overall pattern is that the inherent pacemaking site is preferentially located near the outflow end of the vessel, producing retrograde contraction waves, and that any shifts in the direction of conduction tend to be determined by the segment of the vessel experiencing the highest pressure. This conclusion is consistent with the results of previous studies (Bertram et al., 2018; Castorena-Gonzalez et al., 2020), but the present results are more definitive because in the earlier studies there was no evidence of intact NO signalling, which was unlikely in the case of mouse vessels due to the lack of ʟ-arginine supplementation to compensate for the necessity of using relatively high-resistance pipettes.

## Physiological relevance

Our imposed flow protocol aimed to simulate conditions experienced by mesenteric collectors during/after postprandial increases in absorption by the lacteal lymphatic capillaries. Under those conditions the dilatation of collecting vessels and cessation of their spontaneous

contractions lower their resistance to flow (Meisner et al., 2007; Quick et al., 2009), optimizing lymph transport when a favourable pressure gradient is present. Most evidence supports a major role for NO in this process (Gasheva et al., 2006, 2013), but some studies also find a role for prostanoids (DuToit et al., 2024). In the present study, imposed flow was used primarily as a test to confirm NO production, and our focus was on the extent to which NO produced by the pulsatile flow/SS associated with spontaneous contractions modulates lymphatic contractile parameters, including the site of pacemaking initiation and the direction of contraction wave conduction. Because relatively long isolated lymphatic segments were required for our protocols (in which pipette and vessel resistance limited the absolute flow/SS achievable), L-arginine supplementation was needed to enhance NO bioavailability. However even in the absence of L-arginine, NO bioavailability was not impaired, as illustrated by the responsiveness to ACh and the consistent inhibition by flow of contraction frequency in shorter, two-valve segments. Had we not used L-arginine supplementation in studies with longer vessel segments, we would be uncertain whether sufficient NO was produced to test hypotheses about the influence of NO on the pacemaking site and conduction direction. Our results show that although pulsatile SS does indeed modulate contraction frequency and amplitude, it does *not* control the pacemaking site or conduction direction, and this conclusion holds even when SS-induced NO production is enhanced by L-arginine supplementation.

## Factors that determine the LMC pacemaking site *ex vivo*

The observation that collecting vessel segments can be progressively shortened and still develop spontaneous, entrained contractions *ex vivo* (Davis & Zawieja, 2025; van Helden, 1993; Zawieja, Castorena-Gonzalez, To et al., 2018) suggests that any or most LMC(s) can become the pacemaking initiation site that then drives entrained action potentials through a contiguous network of connexin-coupled LMCs and thereby triggers co-ordinated contraction waves of the entire vessel segment. A previous modelling study predicted that LMCs at the upstream side of a valve sinus might be preferential pacemakers (Hald et al., 2018), not because they represent a specialized LMC subpopulation but due to a lower degree of connexin-mediated coupling to adjacent LMCs, which are present at reduced density in the sinus region (see references cited by Davis et al., 2024). The higher distensibility of the sinus (Bertram & Davis, 2023) may also render LMCs in that region more mechanosensitive. *Ex vivo* vessel preparations tend to be more entrained

than *in vivo* ones (Castorena-Gonzalez et al., 2018; Crowe et al., 1997; Liao et al., 2011; Zawieja et al., 1993), presumably because the severed vessel ends of the former reduce electrical current sink (Hald et al., 2018). Another effect of cutting the vessel ends, along with possible LMC damage from sutures needed to secure the vessel to glass micropipettes, could be to promote the development of artefactual pacemaking sites that override an intrinsic pacemaker. However if artefactual pacemakers were introduced at both ends by our procedures, we would have expected to observe a 50:50 retrograde–antegrade conduction wave mixture (due to equivalent trauma at both vessel ends), whereas we nearly always observed the same ∼70:30 retrograde-antegrade ratio (Figs 3*B*, 4*B*, 6*B*) documented *ex vivo* (Castorena-Gonzalez et al., 2020). Considering all these factors our data support previous models (Bertram et al., 2018; Hald et al., 2018) in which the microenvironment of the vessel wall and the endogenous mechanical forces (primarily pressure/stretch) at a particular site dictate which LMC(s) become(s) the default pacemaker for the vessel. NO produced by pulsatile SS plays little, if any, role in determining the pacemaking site, but greater NO production associated with imposed flow (i.e. steady SS) attenuates the conduction of the contraction wave (Figs. 3*D*,*E* and 6*E*) or interrupts it at susceptible points along the vessel, where it may or may not regenerate (Fig. 4*E*). A possible mechanism for the attenuation of conduction by imposed flow is that the further a wave conducts from the origination site, the more it may gradually attenuate, with any additional inhibition of ionic conductance by flow-mediated NO production enhancing that attenuation.

Multiple models have examined how the combined effects of flow/SS and pressure impact lymphatic pumping (Bertram et al., 2019; Caulk et al., 2016; Contarino & Toro, 2018; Quick et al., 2007; Razavi et al., 2020; Sedaghati et al., 2023). Others that considered conduction took the approach of imposing a direction and rate and then examined the consequences for pumping (Bertram et al., 2011, 2016; Elich et al., 2021; Sedaghati et al., 2023; Venugopal et al., 2007; Wolf et al., 2023). However only three models (Bertram et al., 2018; Kunert et al., 2015; Li et al., 2022) have made testable predictions about the direction of contraction wave conduction and/or pacemaking site. Our experimental observations do not support the concept that fluctuations in NO production can, in themselves, trigger contractions, thus contradicting predictions made by the models of Kunert et al. (2015) and Baish et al. (2016). Although those models cite experimental support from studies showing entrainment of spontaneous contractions to oscillatory imposed flow (Kornuta et al., 2015; Mukherjee et al., 2019), the mechanism of that particular behaviour was shown to be

independent of NO. Likewise any inherent assumption that NO production is required to terminate contractions would be contradicted by the persistence of spontaneous contractions with normal diastolic relaxations in rat vessels treated by L-NAME and without L-arginine supplementation (Gashev et al., 2002, 2004, 2006; present results), as well as in mouse vessels after complete genetic deletion of eNOS (Scallan & Davis, 2013). The inaccurate predictions of these numerical models point to a need for better ones that explain the interactions of SS and pressure while also matching the experimental observations of this and previous studies.

A starting point for an improved numerical model might be a combination of those developed by Bertram et al. (2018, 2019), one of which focused on the effects of selective $P_{out}$ elevation but also incorporated realistic valve behaviour, and the other which focused on SS-induced effects on amplitude and frequency. These two models operate in distinctly different physiological domains: the former applies to the low pulsatile flows induced by lymphangion contractions against an adverse pressure difference, whereas the latter applies to the high-steady-flow conduit regime where all the valves are open and contractions are inhibited. Consequently only the 2018 model makes predictions about the direction of contraction wave conduction. A contraction of any lymphangion causes activation signals to travel to both adjacent lymphangions at a finite rate, where such signals can initiate a local contraction if these neighbouring lymphangions are not temporarily refractory because of a recent or ongoing contraction. This last provision prevents contraction waves doubling back on themselves under normal circumstances. The model emulates observed behaviour in that the direction of contraction wave conduction can be controlled by manipulating the inlet/outlet pressure difference (Bertram et al., 2018) and is capable of arrhythmic contraction patterns in the absence of noise perturbations. Bertram (2024) modified the model to allow for lymphangions of varying length and in a supplement showed many additional complex interactions between lymphangions, both periodic and aperiodic. Thus far motivation to add SS-induced effects to the model has been lacking in the absence of unambiguous evidence that the domain where this is important overlaps with the pumping regime. The present results show that SS-induced effects have only minimal influence on pacemaking initiation site and conduction direction when the lymphatic vessel is in pumping mode, but some influence on contraction frequency and amplitude (assuming that the differences we and Gasheva et al. (2006) show are not caused by off-target effects of L-NAME). More accurate models would need to reflect this nuanced situation. The effects of SS-induced prostanoid production could also be incorporated into future models.

## Implications for disease and/or therapy

The clinical relevance of our findings relates to the largely unknown mechanisms underlying the impaired lymph flow and pumping observed in human extremities during lymphoedema and/or inflammation. Multiple studies point to impaired lymphatic contractile function under these conditions (Davis, Castorena-Gonzalez et al., 2023; Davis, Scallen et al., 2023; DuToit et al., 2024; Liao et al., 2011; Mathias & von der Weid, 2013; Olszewski, 2002; Scallan et al., 2021; Zawieja et al., 2016), and it will be important to know the extent to which SS- and/or pressure-mediated mechanisms might be targeted therapeutically to improve lymphatic contractile function and lymph transport.

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

## Additional information

### Data availability statement

The data that support the findings of this study are available from the corresponding author upon reasonable request.

### Competing interests

The authors declare that they have no competing interests.

### Author contributions

M.J.D. and C.D.B. designed the study. M.J.D. conducted the experiments. M.J.D. and C.D.B. analysed the data. M.J.D. prepared the figures and drafted the manuscript. M.J.D. and C.D.B. edited the manuscript. Both authors have read and approved the final version of this manuscript and agreed to be accountable for all aspects of the work, ensuring that questions related to accuracy or integrity of any part of the work are appropriately investigated and resolved. All persons designated as authors qualify for authorship and all who qualify for authorship are listed.

### Funding

This work was supported by NIH grants R01 HL-125608 and R01-HL-122578 to M.J.D.

### Acknowledgements

We are grateful to Dr. Jorge Castorena-Gonzalez of Tulane University for supplying his software LySS_contraSTM_Spectro to create the spatiotemporal maps.

### Keywords

action potential conduction, contraction wave, imposed flow, nitric oxide, pulsatile shear stress

### Supporting information

Additional supporting information can be found online in the Supporting Information section at the end of the HTML view of the article. Supporting information files available:

**Peer Review History**

