## [Peer Review History · The Journal of Physiology]

Control of lymphatic pacemaking and pumping by mechanobiological signals

Michael J Davis and Christopher D Bertram

DOI: 10.1113/JP288477

Corresponding author(s): Michael Davis (davismj@health.missouri.edu)

Review Timeline:

Submission Date:	09-Jan-2025
Editorial Decision:	30-Jan-2025
Revision Received:	16-Apr-2025
Accepted:	23-Apr-2025

Senior Editor: Kim Barrett

Reviewing Editor: Bernard Drumm

Transaction Report:

Dear Dr Davis,

Re: JP-RP-2025-288477 "Control of lymphatic pacemaking and pumping by mechanobiological signals" by Michael J Davis and Christopher D Bertram

Thank you for submitting your manuscript to The Journal of Physiology. It has been assessed by a Reviewing Editor and by 2 expert referees and we are pleased to tell you that it is potentially acceptable for publication following satisfactory major revision.

REVISION CHECKLIST:

Please upload two versions of your manuscript text: one with all relevant changes highlighted and one clean version with no

changes tracked. The manuscript file should include all tables and figure legends, but each figure/graph should be uploaded as separate, high-resolution files.

We look forward to receiving your revised submission.

Yours sincerely,

Kim Barrett
Senior Editor
The Journal of Physiology

REQUIRED ITEMS

- Author photo and profile. First or joint first authors are asked to provide a short biography (no more than 100 words for one author or 150 words in total for joint first authors) and a portrait photograph. These should be uploaded and clearly labelled together in a Word document with the revised version of the manuscript. See Information for Authors for further details.

- You must start the Methods section with a paragraph headed Ethical approval (https://jp.msubmit.net/cgi-bin/main.plex?form_type=display_requirements#methods).

Research must comply with The Journal's policies regarding animal experiments (<https://physoc.onlinelibrary.wiley.com/hub/animal-experiments>) and adherence to these policies must be stated in the manuscript.

Authors should confirm in their Methods section that their experiments were carried out according to the guidelines laid down by their institution's animal welfare committee, including an ethics approval reference number. The Methods section must contain a statement about access to food, water and housing, details of the anaesthetic regime: anaesthetic used, dose and route of administration, and method of killing the experimental animals.

- The Journal of Physiology funds authors of provisionally accepted papers to use the premium BioRender site to create high resolution schematic figures. Follow this link and enter your details and the manuscript number to create and download figures. Upload these as the figure files for your revised submission. If you choose not to take up this offer, we require figures to be of similar quality and resolution. If you are opting out of this service to authors, state this in the Comments section on the Detailed Information page of the submission form. The link provided should only be used for the purposes of this submission. Authors will be charged for figures created on this premium BioRender account if they are not related to this manuscript submission.

- Please upload separate high-quality figure files via the submission form.

- Please ensure that the Article File you upload is a Word file.

- Papers must comply with the Statistics Policy: https://jp.msubmit.net/cgi-bin/main.plex?form_type=display_requirements#statistics.

In summary:

- If $n \leq 30$, all data points must be plotted in the figure in a way that reveals their range and distribution. A bar graph with data points overlaid, a box and whisker plot or a violin plot (preferably with data points included) are acceptable formats.

- If $n > 30$, then the entire raw dataset must be made available either as supporting information, or hosted on a not-for-profit repository, e.g. FigShare, with access details provided in the manuscript.

- 'n' clearly defined (e.g. x cells from y slices in z animals) in the Methods. Authors should be mindful of pseudoreplication.

- All relevant 'n' values must be clearly stated in the main text, figures and tables.

- The most appropriate summary statistic (e.g. mean or median and standard deviation) must be used. Standard Error of the Mean (SEM) alone is not permitted.

- Exact p values must be stated. Authors must not use 'greater than' or 'less than'. Exact p values must be stated to three significant figures even when 'no statistical significance' is claimed.

- Please include an Abstract Figure file, as well as the Figure Legend text within the main article file. The Abstract Figure is a piece of artwork designed to give readers an immediate understanding of the research and should summarise the main conclusions. If possible, the image should be easily 'readable' from left to right or top to bottom. It should show the physiological relevance of the manuscript so readers can assess the importance and content of its findings. Abstract Figures should not merely recapitulate other figures in the manuscript. Please try to keep the diagram as simple as possible and without superfluous information that may distract from the main conclusion(s). Abstract Figures must be provided by authors no later than the revised manuscript stage and should be uploaded as a separate file during online submission labelled as File Type 'Abstract Figure'. Please also ensure that you include the figure legend in the main article file. All Abstract Figures should be created using BioRender. Authors should use The Journal's premium BioRender account to export high-resolution images. Details on how to use and access the premium account are included as part of this email.

Reviewing Editor's comments:

Thank you for your submission to the Journal.

Both referees have commended the high standard of technical expertise displayed in this study, with both pointing out the difficulty in acquiring such recordings from these vessels *ex vivo*. The referees have raised some important points that need to be addressed however before further consideration.

In particular, the concerns of reviewer 2 address the physiological relevance of the phenomenon under scrutiny. The

reviewer is concerned on the lack of flow-induced inhibition of contractions unless the vessels were supplemented with 1mM L-Arg. The issue of why NO bioavailability is comprised and supposed NO mediated are absent in the majority of ex-vivo preparations needs careful discussion (at a minimum), as currently the lack of inhibitory effects from flow without high concentrations of L-Arg make the physiological relevance of these observations somewhat challenging to deduce. The referee also points to issues surrounding vessel viability that requires careful consideration.

Also please carefully review the Journal's policies on clearly stating use of Standard Deviation for summary data, as well as animal access to food and water.

Senior Editor:

Comments to ensure the paper complies with the Statistics Policy:

Please see comments from the RE

Referee #1:

Comments to the authors:

The manuscript by Davis and Bertram is reporting attempts to experimentally test and challenge previously published modelling predictions related to the influence of pressure and flow/shear stress (SS) on lymphatic vessel pacemaking and pumping behaviour. It provides in an elegant way a better understanding of the regulation of the active lymphatic pump by pressure and flow.

The manuscript is well-written, and the objectives are extremely clearly defined and detailed. The experimental design has been well planned and organised, the methodology is perfectly mastered and the data interpretation sound, leading to solid and well supported conclusions.

I only have minor comments and suggestions that I have listed below.

1. One of the first observation made by the authors was an inconsistent vessel response to SS that they interpreted as caused by a diminished NO production in their ex vivo experimental conditions and thus considered NO as the only endothelial determinant of SS/flow effect on lymphatic pumping. It should be noted that several other studies investigating lymphatic responses to flow in similar ex vivo systems have reported a contribution of both contractile and dilatory

prostanoids to SS-induced responses which may antagonistically act to minimize the pumping inhibition. While the authors' interpretation of a major role for NO seems to be confirmed by the restoration of the inhibitory effect of SS upon addition of L-Arginine to the luminal perfusing solution, the lack of inhibition of that response with L-NAME may point to an enhanced role of prostanoids. This comment may have little bearings in the context of the current study, but it might be important to discuss the potential role of prostanoids in the contractile response to flow and consider it as potential limitation. Interestingly, one of these studies where the role of prostanoids is implicated also reported flow-responsive and -unresponsive vessels in a similar experimental system (DuToit et al., 2024, <https://doi.org/10.1111/micc.12839>).

2. The authors mentioned line 154 that; "Both male and female rats were used but no attempt was made to analyze the results by sex". It is surprising, given that experiments have been performed, that sex-dependent analysis has not been considered, as it may reveal interesting differences.

3. Line 173: PSS is mentioned here but Krebs solution seems to be the buffered saline solution used. Is it a typo?

4. Line 655: in this Fig 2 caption, detected edges are said to be identified by yellow dots and the line of best fit be in red. Either they are not there, or the very small size of the figure makes it impossible to see.

5. Lines 700-704: To align with the other figure captions (Fig 3 and 4), description is not necessary here and duplicates account in the Results section. Additionally, description in B seems wrong/inverse.

Referee #2:

This is an interesting and well-conducted study of pumping in rat mesenteric lymphatics. The authors are to be applauded on their immense technical ability to produce beautiful records of lymphatic activity from these preparations, given that the walls of these vessels are just a few cells thick.

Nevertheless, I have a number of concerns about the paper especially since the vast majority (1/13) of preparations did not show flow-induced inhibition of contractions, unless solutions were supplemented with 1mM L-ARG and (in some cases) 100 uM N hydroxy-nor-L-arginine.

1. The authors use the term "control vessels" frequently (eg L532). However, I think it would be more accurate to use the term L-ARG-treated vessels (or something similar) so that the reader is absolutely clear that preparations were recorded after trying to boost NO production.

2. If flow-induced inhibition of contractions is so difficult to demonstrate in this preparation (& also in the mouse) is it physiologically important?

3. In relation to the above point, would it be worth checking if the endothelial layer in these preparations remains intact?

4. I have a real concern about the validity of the L-NAME experiments, given that only 1/13 untreated preparations showed any flow-induced inhibition. I think that paired controls with perhaps a lower concentration of L-ARG before and after L-NAME is crucial to make this point.

5. The authors have labelled Pin and Pout inconsistently (eg the Pin label in the top panels of Fig1A-C is in red text, but Pout in the middle Pressure trace is shown in red. The authors should consider swapping the colour of the Pin pressure trace to red. I think this will also add clarity to the "Position" STM traces as a red line will signify propagation of the contractile wave in the retrograde direction towards the Pin pipette.

6. As mentioned, the quality of recordings presented is excellent, but I do have a concern about records such as that shown in Figure 3D. Unfortunately, some many lymphatic preparations fail to maintain robust contractions for the duration of the experiment and I think that may obscure some of the findings. The record shown in Figure 3D appears typical of a non-robust preparation and I suspect that its failure to show any recovery should have excluded it from the analysis. The contrast between Figure 3D and practically every other record in the manuscript is quite sharp. I would suggest that the authors

examine the possibility of excluding any lymphatic that didn't recover from/survive the experimental control on their conclusions. The authors may decide what cutoff to use to signify "recovery" is sufficient.

7. In relation to the above point, in L325-L330, how do you know that the lymphatic simply hadn't died?

8. Please state in which experiments 100 uM N hydroxy-nor-L-arginine was used in addition to L-NAME.

Minor Points:

1. Please provide a scale bar on all images of lymphatics.

2. Have the authors any concerns about the effect of xylazine (administered as anaesthetic) on lymphatic function? Could this alter function of the endothelial cells? Have the authors any data on this?

3. L281- please state the percentages of preparations in which contractions initiated centrally.

4. L463 The discussion is interesting but I think it is too long and needs to be significantly shortened. I'm not sure if L463-L489 are particularly relevant to this manuscript.

END OF COMMENTS

Reviewing Editor's comments:

Thank you for your submission to the Journal.

Both referees have commended the high standard of technical expertise displayed in this study, with both pointing out the difficulty in acquiring such recordings from these vessels *ex vivo*. The referees have raised some important points that need to be addressed however before further consideration.

In particular, the concerns of reviewer 2 address the physiological relevance of the phenomenon under scrutiny. The reviewer is concerned on the lack of flow-induced inhibition of contractions unless the vessels were supplemented with 1mM L-Arg. The issue of why NO bioavailability is comprised and supposed NO mediated are absent in the majority of *ex-vivo* preparations needs careful discussion (at a minimum), as currently the lack of inhibitory effects from flow without high concentrations of L-Arg make the physiological relevance of these observations somewhat challenging to deduce.

- In retrospect we can understand how our finding that flow-induced inhibition of phasic lymphatic contractions occurred in only 1 of 13 vessels without L-arginine supplementation, together with various related statements in the Abstract, Results, and Discussion describing that result, could be interpreted as a vessel viability problem. However, it is not. Rather, it is a limitation on the level of flow/SS that can be imposed in the longer vessel segments required for our experiments. We explain why in detail in our response to Reviewer 2 and have performed additional experiments using shorter 2-valve vessel segments to prove our point.

The referee also points to issues surrounding vessel viability that requires careful consideration.

- To determine whether the endothelium of our preparations was intact and able to produce nitric oxide (NO) in the *absence* of L-arginine supplementation, we performed additional experiments testing vessel responsiveness to the EC-dependent dilator ACh. Phasic contractions of all (11 of 11) our vessels consistently decreased in frequency (by an average of ~30%) in response to 1 or 10 μ M ACh and that inhibition was blocked by L-NAME (100 μ M). In our response to Reviewer 2 we show an example of the ACh protocol, the summary data and the associated statistical tests. These results indicate that the endothelium of our *ex-vivo* mesenteric lymphatic segments can produce NO in the *absence* of L-arginine supplementation.

[T]he concerns of reviewer 2 address the physiological relevance of the phenomenon under scrutiny.

- The physiological relevance of flow-induced inhibition of lymphatic contractions is not in question, as multiple *in-vivo* studies (cited in our manuscript) have shown that the transition of actively pumping collecting vessels to passive conduits lowers their resistance to flow (Quick et al. 2007, 2009), optimizing lymph transport when a favorable pressure gradient for flow is present. Importantly, cessation of lymphatic contractions occurs in response to flow in mesenteric lymphatic collectors during/after post-prandial increases in absorption by the lacteal lymphatic capillaries (Lee 1979). Further, the most relevant studies show that this transition involves SS-induced NO production (Gasheva 2006, 2013), although other studies also implicate prostanoid production. The relevant issue in our study concerns whether NO production from the pulsatile flow/ SS associated with spontaneous contractions significantly modulates lymphatic contractile parameters, including the site of pacemaking initiation and the direction of contraction wave

conduction, as predicted by a widely accepted numerical model of lymphatic behavior. Our results show that, while pulsatile SS does indeed modulate contraction frequency and amplitude, it does NOT control the pacemaking site or conduction direction, and this conclusion holds even when SS-induced NO production is enhanced by L-arginine supplementation. As we now show, L-arginine supplementation was required *not to rescue impaired NO bioavailability* but to enhance it in the relatively long isolated lymphatic segments (containing 2-3 complete lymphangions) required for our experiments, where pipette and vessel resistance limited the absolute flow/SS. We discuss the resistance issues extensively in our response to Reviewer 2, as well as describing the additional experiments performed to substantiate our argument. If we had not used L-arginine supplementation with the longer vessel segments, it could be argued that insufficient NO was produced under the conditions of our experiments to be able to test hypotheses about the influence of NO on the pacemaking site and conduction direction.

Also please carefully review the Journal's policies on clearly stating use of Standard Deviation for summary data, as well as animal access to food and water.

- Done. The use of SD is now clearly stated in the statistics section (L312) and we have replotted all relevant panels of each figure using SD.

Referee #1:

Comments to the authors:

The manuscript by Davis and Bertram is reporting attempts to experimentally test and challenge previously published modelling predictions related to the influence of pressure and flow/shear stress (SS) on lymphatic vessel pacemaking and pumping behaviour. It provides in an elegant way a better understanding of the regulation of the active lymphatic pump by pressure and flow.

The manuscript is well-written, and the objectives are extremely clearly defined and detailed. The experimental design has been well planned and organised, the methodology is perfectly mastered and the data interpretation sound, leading to solid and well supported conclusions.

- Thank you.

I only have minor comments and suggestions that I have listed below.

1. One of the first observation made by the authors was an inconsistent vessel response to SS that they interpreted as caused by a diminished NO production in their ex vivo experimental conditions and thus considered NO as the only endothelial determinant of SS/flow effect on lymphatic pumping.

- When we used the phrase “enhanced NO bioavailability” we did not mean to imply that our preparations had impaired NO bioavailability in the absence of L-arginine. Indeed, that is not the case as we now demonstrate with additional experimental data collected in response to issues raised by Reviewer 2 (points 2 and 4). The new experiments show that we observe flow-induced responses comparable to those of DuToit et al. (2024) *in the absence of L-arginine supplementation* when we use shorter, 2-valve mesenteric lymphatic segments, confirming that the apparent flow unresponsiveness of 3-4 valve segments is likely related to higher resistance to flow rather than

compromised vessels or impaired NO responsiveness. Please see our response to points 2 and 4 of Reviewer 2.

It should be noted that several other studies investigating lymphatic responses to flow in similar ex vivo systems have reported a contribution of both contractile and dilatory prostanoids to SS-induced responses which may antagonistically act to minimize the pumping inhibition. While the authors' interpretation of a major role for NO seems to be confirmed by the restoration of the inhibitory effect of SS upon addition of L-Arginine to the luminal perfusing solution, the lack of inhibition of that response with L-NAME may point to an enhanced role of prostanoids. This comment may have little bearings in the context of the current study, but it might be important to discuss the potential role of prostanoids in the contractile response to flow and consider it as potential limitation. Interestingly, one of these studies where the role of prostanoids is implicated also reported flow-responsive and -unresponsive vessels in a similar experimental system (DuToit et al., 2024, <https://doi.org/10.1111/micc.12839>).

- We agree that the data of DuToit et al. (2024) provide support for both prostanoids and NO production by LECs in response to flow/SS. Several studies used a combination of NO and prostanoid inhibitors together to block flow-induced frequency inhibition but did not distinguish between components of flow-induced frequency inhibition by NO or other products (Gashev et al. 2002, Gashev et al. 2004). We have now cited that study a number of times in the revised manuscript. The data of DuToit et al. (2024) are exceptional as those authors used L-NNA and indomethacin separately. Their finding that *either* NO or prostanoid inhibition completely blocked flow-induced effects is intriguing as it implies that the two pathways are activated sequentially (but that unconventional possibility was not discussed in their paper). As you note, the focus of our study was limited to NO-mediated responses, in large measure because the hypotheses and models of Kunert et al. (2015) and Baish et al. (2016) focused exclusively on NO and ignored non-NO components. SS-mediated prostanoid production may well explain our inability to effectively block SS-mediated inhibition of phasic contractions using L-NAME. We have now mentioned this possibility and discussed the possible roles of prostanoids (L572-574 in Discussion). We also suggest that SS-induced prostanoid production and its effects need to be incorporated into future numerical models of lymphatic behavior (line 655).

2. The authors mentioned line 154 that; "Both male and female rats were used but no attempt was made to analyze the results by sex". It is surprising, given that experiments have been performed, that sex-dependent analysis has not been considered, as it may reveal interesting differences.

- We agree, but adding this additional element to our experimental protocol would have potentially required doubling the number of animals needed and the number of figure panels needed to present the results. Although that information would be valuable, it would detract from the central focus of our study. We note, however, that preliminary analyses of the responses of our vessels to flow, Pin elevation and Pout elevation did not reveal any striking sex differences, reinforcing our results from a previous study (PMID 36714313) that found no significant differences between normal male and female animals in several aspects of contractile function.

3. Line 173: PSS is mentioned here but Krebs solution seems to be the buffered saline solution used. Is it a typo?

- Fixed.

4. Line 655: in this Fig 2 caption, detected edges are said to be identified by yellow dots and the line of best fit be in red. Either they are not there, or the very small size of the figure makes it impossible to see.

- Thank you. We have now enlarged that image and show it as panel B of Fig. 2.

5. Lines 700-704: To align with the other figure captions (Fig 3 and 4), description is not necessary here and duplicates account in the Results section.

- Fixed, lines 735-737.

Additionally, description in B seems wrong/inverse.

- Fixed.

Referee #2:

This is an interesting and well-conducted study of pumping in rat mesenteric lymphatics. The authors are to be applauded on their immense technical ability to produce beautiful records of lymphatic activity from these preparations, given that the walls of these vessels are just a few cells thick.

- Thank you.

Nevertheless, I have a number of concerns about the paper especially since the vast majority (1/13) of preparations did not show flow-induced inhibition of contractions, unless solutions were supplemented with 1mM L-ARG and (in some cases) 100 uM N hydroxy-nor-L-arginine.

- We address these concerns in detail below (see response to point 4).

1. The authors use the term "control vessels" frequently (eg L532). However, I think it would be more accurate to use the term L-ARG-treated vessels (or something similar) so that the reader is absolutely clear that preparations were recorded after trying to boost NO production.

- Point taken — more precise language has been substituted.

2. If flow-induced inhibition of contractions is so difficult to demonstrate in this preparation (& also in the mouse) is it physiologically important?

- Yes, evidence for a physiologically important role comes from the in-vivo studies of Lee (1979) who showed that the active pumping of mesenteric collectors is inhibited post-prandially, and from studies by Quick et al. (2007, 2009) who showed that the switch of lymphatic collectors from pumps to conduits facilitates rather than impairs lymph transport under these conditions (all studies that we have cited). In our present study, the primary purpose of testing the effects of imposed flow was to ensure sufficient NO bioavailability that any NO associated with pulsatile SS could *potentially* exert its effects on contraction amplitude, frequency, conduction direction, etc. under the conditions of our experiments (now clearly stated in L574-577). However, NO produced by pulsatile SS did NOT alter the pacemaking site or conduction direction, *with or without* enhancement of NO bioavailability by L-arginine.

- This concern is addressed in more detail under point 4 and we have now added a short paragraph to the Discussion addressing some of these issues (after first deleting a more-than-equivalent amount of text).

3. In relation to the above point, would it be worth checking if the endothelial layer in these preparations remains intact?

- Yes, although our observation that 3-4 valve vessels respond strongly to flow after L-arginine supplementation is itself confirmation that the endothelial layer was intact. An important issue is whether *normal* NO bioavailability was impaired in our preparations. To test endothelial cell viability in the *absence* of L-arginine supplementation, we performed an additional series of experiments on isolated rat mesenteric lymphatic vessels containing either 2 valves or 3-4 valves (for explanation see below). The results from vessels with different numbers of valves were similar, so we combined them.

- 11 out of 11 vessels were responsive to ACh. The example on the left illustrates the protocol. After the vessel developed spontaneous contractions, we tested 1 μM and 10 μM ACh and averaged the peak response to each over 30-60 sec; both concentrations were tested again after incubating the vessel in L-NAME (100 μM) for 20-30 min. The summary data show ~30% decrease in contraction frequency that is blocked by L-NAME. Thus, 100% of our isolated lymphatic preparations have an intact endothelium and intact NO signaling in the *absence* of L-arginine supplementation, i.e. *without* the need to enhance NO bioavailability. We felt that adding an additional figure showing these data would detract from the central focus of the manuscript, but we have now cited these results in the text of the Results (L-328-334).

4. I have a real concern about the validity of the L-NAME experiments, given that only 1/13 untreated preparations showed any flow-induced inhibition.

- In retrospect we can understand how our finding that flow-induced inhibition of phasic contractions occurred in only 1 out of 13 of our 3-4 valve vessels without L-arginine supplementation could be interpreted as a problem with vessel viability or compromised NO

bioavailability. However, it is not; rather, it is a limitation related to the level of SS that can be imposed in the longer vessel segments required for our experiments. We explain why below and have performed additional experiments to demonstrate our points.

- There are at least two reasons why only 1 out of 13 of our 3-4 valve vessels without L-arginine supplementation responded to flow, as compared to 82% flow-responsive vessels in the recent study by DuToit et al., 2024 ([https://doi.org/ 10.1111/micc.12839](https://doi.org/10.1111/micc.12839); cited by Reviewer 1). [1] DuToit et al. also studied rat mesenteric lymphatic collectors ex vivo but used *shorter* vessels, each containing a single lymphangion, whereas we used vessels with 2-3 complete lymphangions; the longer lengths are predicted to result in 2-3 times lower absolute flows / shear stresses when flow is imposed at a given trans-axial pressure gradient (ΔP). Longer vessel segments were required for our protocols to accurately measure conduction speed and direction and to determine whether the various interventions disrupted entrainment between coupled lymphangions. [2] The maximal ΔP used to impose flow was 4 cmH₂O in our study (Pin = 5, Pout = 1 cmH₂O), compared to 8 cmH₂O used by DuToit et al.; this would result in a further reduction in the relative flows / shear stresses experienced by our vessels. Taken together, these two variables would equate to approximately **4-6× lower flows and shear stresses** during our imposed flow protocol in longer vessel segments. [Pipette sizes/resistances are also critical but those were not reported by DuToit et al. (2024) and so no relevant comparisons of pipette resistances to ours can be made.] Thus, the relative lack of responsiveness is likely due to the lower imposed flow/SS rather than to compromised vessels.

- To illustrate these points, we performed a new set of experiments using rat mesenteric lymphatic vessels containing only 2 valves, similar to the segment lengths used by DuToit et al. (2024), while using the same pair of cannulating pipettes and the same imposed ΔP as in our earlier experiments, but in the **absence of L-arginine supplementation**.

- Under these conditions 5 out of 6 two-valve vessels were responsive to flow. (We also tested four additional 3-4 valve vessels, and 0 out of 4 were responsive to flow in the absence of L-arginine supplementation, confirming our previous results.) The example and summary data above show that we get a very similar degree of flow-induced inhibition of contraction frequency as DuToit et al. 2024 (54% of control vs ~55-60% in Fig. 1C of their paper), and the flow response occurs even though we used a maximal $\Delta P = 4$ cmH₂O as compared to $\Delta P = 8$ cmH₂O in their study. Therefore, our vessels are comparably or even slightly more responsive to flow than theirs. Additionally, the response to flow was blocked by L-NAME, similar to their findings. Again, we felt that adding an

additional figure showing these data would detract from the central focus of the manuscript, but we have now cited these results in the text of the Results (L-334-342).

I think that paired controls with perhaps a lower concentration of L-ARG before and after L-NAME is crucial to make this point.

- We agree that a paired experimental design would be better, and in initial experiments we attempted this—the use of lower concentrations of L-arginine that would enable flow-induced responses in 3-4 valve lymphatic segments but still be potentially blocked (in the same vessel) by L-NAME. It turned out to be extremely difficult, requiring a new vessel and different pipettes/solutions to test each concentration of L-arginine. Even given this extra effort, the effects were highly variable, e.g. 0.1 mM L-arginine was sufficient for some but not most vessels. The degree of blockade by L-NAME, even with only 0.1 mM L-arginine, was also highly variable. In retrospect, we think that part of the issue is that both NO and prostanooids are produced in response to imposed flow, as raised by Reviewer 1 and found by DuToit et al. (2024). We now address this issue in the Discussion (L573-574 and L658-659).

5. The authors have labelled Pin and Pout inconsistently (eg the Pin label in the top panels of Fig1A-C is in red text, but Pout in the middle Pressure trace is shown in red. The authors should consider swapping the colour of the Pin pressure trace to red. I think this will also add clarity to the "Position" STM traces as a red line will signify propagation of the contractile wave in the retrograde direction towards the Pin pipette.

- Thank you for spotting this inconsistency. We have made these corrections. However, changing all Pin traces from blue to red would be the opposite convention to that used in some 20 of our previous papers. With respect to the line colors used to code direction in STMs, red indicates the site of the pacemaker being near the Pout pipette (and conducting away from that area), and blue the opposite, as now stated (lines 688-689).

6. As mentioned, the quality of recordings presented is excellent, but I do have a concern about records such as that shown in Figure 3D. Unfortunately, some many lymphatic preparations fail to maintain robust contractions for the duration of the experiment and I think that may obscure some of the findings. The record shown in Figure 3D appears typical of a non-robust preparation and I suspect that its failure to show any recovery should have excluded it from the analysis. The contrast between Figure 3D and practically every other record in the manuscript is quite sharp. I would suggest that the authors examine the possibility of excluding any lymphatic that didn't recover from/survive the experimental control on their conclusions. The authors may decide what cutoff to use to signify "recovery" is sufficient.

- We did in fact terminate any experiment if a vessel failed to recover a contraction amplitude of at least 75% of control after one of the protocols, and this is now stated in the Methods. However, recovery times from the imposed flow protocol were the most variable, and not all vessels recovered within 2 min after imposed flow ceased; in some the inhibition of frequency lasted, but most eventually resumed spontaneous contractions. The original example in Fig. 3 was one such case, in which the vessel required an additional 6 min to resume contractions (18 contractions occurred before the subsequent protocol was started). Nevertheless, to avoid confusion we have substituted a different example for Fig. 3D.

7. In relation to the above point, in L325-L330, how do you know that the lymphatic simply hadn't died?

- Please see our response to point 6.

8. Please state in which experiments 100 uM N hydroxy-nor-L-arginine was used in addition to L-NAME.

- Done, L217-218.

Minor Points:

1. Please provide a scale bar on all images of lymphatics.

- Scale bars have been added to the vessel images in Figures 1-2. Thank you for pointing out this omission.

2. Have the authors any concerns about the effect of xylazine (administered as anaesthetic) on lymphatic function? Could this alter function of the endothelial cells? Have the authors any data on this?

- Both ketamine and xylazine (and most other anesthetic agents) have inhibitory effects on contractile function in vivo (see Bachmann et al., PMID: 30829392). However, these agents are much less likely to affect ex vivo preparations such as ours, which are excised immediately after the animal is anesthetized, removed and placed in a relatively large volume of anesthetic-free Krebs solution for cleaning, cannulation and subsequent experimental protocols, in which the vessel is constantly superfused with fresh Krebs solution for periods lasting up to 3 hours.

3. L281- please state the percentages of preparations in which contractions initiated centrally.

4. L463 The discussion is interesting but I think it is too long and needs to be significantly shortened. I'm not sure if L463-L489 are particularly relevant to this manuscript.

- L463-L489 in the original manuscript have been removed and original lines 610-633 have been condensed to two sentences. We shortened the first paragraph of the Discussion by moving original lines 428-435 to a later paragraph (now beginning at line 617). Some subsequent expansion was necessary to address the concerns of the two reviewers and Editor about the role of prostanoids and the physiological relevance of flow-induced responses (lines 568-587). Overall, the Discussion has been shortened from 3197 to 3138 words.

Dear Professor Davis,

Re: JP-RP-2025-288477R1 "Control of lymphatic pacemaking and pumping by mechanobiological signals" by Michael J Davis and Christopher D Bertram

We are pleased to tell you that your paper has been accepted for publication in The Journal of Physiology.

*** IMPORTANT ***

We only seem to have a single PDF file of the figures, and our publisher will require separate high resolution image files (i.e. one for each figure). Please can you email a new set of figure files to us as soon as possible, to: jp@physoc.org.

For further details, see: https://jp.msubmit.net/cgi-bin/main.plex?form_type=display_requirements#generalfigureguide

Yours sincerely,

Kim Barrett
Senior Editor
The Journal of Physiology

If you would like to receive our 'Research Roundup', a monthly newsletter highlighting the cutting-edge research published in The Physiological Society's family of journals (The Journal of Physiology, Experimental Physiology, Physiological Reports, The Journal of Nutritional Physiology and The Journal of Precision Medicine: Health and Disease), please click this link, fill in your name and email address and select 'Research Roundup':

<https://www.physoc.org/journals-and-media/membernews>

- **TRANSPARENT PEER REVIEW POLICY:** To improve the transparency of its peer review process, The Journal of Physiology publishes online as supporting information the peer review history of all articles accepted for publication. Readers will have access to decision letters, including Editors' comments and referee reports, for each version of the manuscript as well as any author responses to peer review comments. Referees can decide whether or not they wish to be named on the peer review history document.
- You can help your research get the attention it deserves! Check out Wiley's free Promotion Guide for best-practice recommendations for promoting your work at: www.wileyauthors.com/eoo/guide. You can learn more about Wiley Editing Services which offers professional video, design, and writing services to create shareable video abstracts, infographics, conference posters, lay summaries, and research news stories for your research at: www.wileyauthors.com/eoo/promotion.
- **IMPORTANT NOTICE ABOUT OPEN ACCESS:** To assist authors whose funding agencies mandate public access to published research findings sooner than 12 months after publication, The Journal of Physiology allows authors to pay an Open Access (OA) fee to have their papers made freely available immediately on publication.

EDITOR COMMENTS

Reviewing Editor:

Thank you for the work on your revised manuscript. Both reviewers have commended the attention to detail in your responses to their queries and highlighted the rigour and potential impact of this study to the field of lymphatic pacemaking.

REFEREE COMMENTS

Referee #1:

The authors have addressed my concerns to my satisfaction. I have no further comments. Congratulations on a study well conducted.

Referee #2:

The authors have comprehensively addressed each one of my queries.